# Monitoring and assimilation tests with TROPOMI data in the CAMS system: Near-real time total column ozone

Antje Inness[1], Johannes Flemming[1], Klaus-Peter Heue[2], Christophe Lerot[3], Diego Loyola[2], Roberto Ribas[1], Pieter Valks[2], Michel van Roozendael[3], Jian Xu[2] and Walter Zimmer[2]

[1]ECMWF, Shinfield Park, Reading, RG2 9AU, UK
[2]German Aerospace Centre (DLR), Remote Sensing Technology Institute, Oberpfaffenhofen, 82234 Wessling, Germany
[3]BIRA-IASB, Brussels, Belgium

*Correspondence to*: Antje Inness (a.inness@ecmwf.int)

## Abstract

The TROPOspheric Monitoring Instrument (TROPOMI) on board the Sentinel 5 Precursor (S5P) satellite launched in October 2017 yields a wealth of atmospheric composition data, including retrievals of total column ozone (TCO3) that are provided in near-real time (NRT) and off-line. The NRT TCO3 retrievals (V1.0.0- V1.1.2) have been included in the data assimilation system of the Copernicus Atmosphere Monitoring Service (CAMS), and tests to monitor the data and to carry out first assimilation experiments with them have been performed for the period 26 November 2017 to 30 November 2018. The TROPOMI TCO3 data agree to within 2% with the CAMS analysis over large parts of the Globe between 60ºN and 60ºS and also with TCO3 retrievals from the Ozone Monitoring Instrument (OMI) and the Global Ozone Monitoring Experiment-2 (GOME-2) that are routinely assimilated by CAMS. However, the TCO3 NRT data from TROPOMI show some retrieval anomalies at high latitudes, at low solar elevations and over snow/ice (e.g. Antarctica and snow-covered land areas in the Northern Hemisphere) where the differences with the CAMS analysis and the other datasets are larger. These differences are particularly pronounced over land in the NH during winter and spring (when they can reach up to 40 DU) and come mainly from the surface albedo climatology that is used in the NRT TROPOMI TCO3 retrieval. This climatology has a coarser horizontal resolution than the TROPOMI TCO3 data which leads to problems in areas where there are large changes in reflectivity from pixel to pixel, e.g. pixels covered by snow/ice or not. The differences between TROPOMI and the CAMS analysis also show some dependency on scan position.

The assimilation of TROPOMI TCO3 has been tested in the CAMS system for data between 60ºN and 60ºS and for solar elevations greater than 10º and is found to have a small positive impact on the ozone analysis compared to Brewer TCO3 data and an improved fit to ozone sondes in the tropical troposphere and to IAGOS aircraft profiles at West African airports. The impact of the TROPOMI data is relatively small because the CAMS analysis is already well constrained by several other ozone retrievals that are routinely assimilated. Averaged over the periods February-April and September-October 2018, difference between experiments with and without assimilation of TROPOMI data are less than 2% for TCO3 and less than 3% in the vertical for seasonal mean zonal mean $O_3$ mixing ratios, with the largest relative differences found in the troposphere.

## 1 Introduction

The Copernicus Atmosphere Monitoring Service (CAMS, atmosphere.copernicus.eu) produces daily global near-real time (NRT) forecasts of atmospheric composition up to five days ahead and a range of other datasets on global and regional atmospheric composition, such as near-real-time estimates of fire emissions, reanalyses of atmospheric composition and

greenhouse gas forecasts and analyses. To improve the quality of the global CAMS forecasts the initial conditions for some of the chemical species, including ozone ($O_3$), carbon monoxide (CO), nitrogen dioxide ($NO_2$), sulphur dioxide ($SO_2$) and for aerosols are improved by assimilating satellite retrievals of atmospheric composition using the 4-dimensional variations (4D-Var) data assimilation system (Benedetti et al., 2009; Inness et al., 2013; Massart et al., 2014, Inness et al., 2015) of the European Centre for Medium Range Weather Forecasts (ECMWF).

A wealth of new atmospheric composition data has become available with the launch of the Sentinel 5-Precursor (S5P) satellite in October 2017. S5P carries the TROPOspheric Monitoring Instrument (TROPOMI) which provides high resolution spectral measurements in the ultraviolet (UV), visible (VIS), near infrared (NIR) and shortwave-infrared (SWIR) part of the spectrum. This wide spectral range allows several atmospheric trace gases to be retrieved, e.g. $O_3$, $NO_2$, $SO_2$ and HCHO from the UVVIS, and CO and $CH_4$ from the SWIR part of the spectrum. These species are all included in the CAMS system, making TROPOMI the perfect instrument to provide observations for the CAMS NRT analysis at unprecedented horizontal resolution of about 3.5 km x 7 km for the TROPOMI UVVIS and 7 km x 7 km for the SWIR products. In this paper, we evaluate TROPOMI NRT total column ozone (TCO3) retrievals (V1.0.0-V1.1.2) produced by the Deutsche Zentrum für Luft- und Raumfahrt (DLR) against the CAMS ozone analysis for the period 26 November 2017 to 30 November 2018 and carry out first assimilation tests with the TROPOMI TCO3 data in the CAMS system. The satellite was still in its so-called commissioning phase until 24 April 2018 which is mainly used for functional testing, in-flight calibration and testing of processing chains. Nevertheless, observations were available in these early months and are included in our study as their quality was good enough to allow us to prepare the CAMS system for the new data.

Ozone plays an important role in tropospheric chemistry. Tropospheric ozone is a regional scale pollutant and, at high concentrations near the surface, harmful to humans and vegetation. Photolysis of ozone, followed by reaction with water vapour, provides the primary source of the hydroxyl radical. Ozone is also a significant greenhouse gas, particularly in the upper troposphere (Hansen et al. 1997). Tropospheric ozone is formed when Nitrogen Oxides (NOx), CO, and Volatile Organic Compounds (VOCs) react in the presence of sunlight. In urban areas in the northern hemisphere (NH) high ozone levels usually occur during spring and summer. About 90% of the total ozone amount resides in the stratosphere, a result of oxygen photolysis as first explained by Chapman (Chapman, 1930). This ozone layer absorbs a large part of the sun's harmful UV radiation. Anthropogenic chlorofluorocarbons led to a global decrease of the ozone total column, with potentially catastrophic consequences avoided thanks to the Montreal Protocol (Newman et al., 2009). Over Antarctica, ozone destruction during Austral spring still leads to strong and rapid depletion of the ozone layer ("ozone hole"). There is evidence that the ozone hole is slowly recovering (Strahan and Douglass, 2018; Weber et al., 2018) and predictions suggest it should return to pre-1980s levels by the 2060s (Newman et al. 2006). Stratospheric ozone destruction also happens on a smaller scale over the Arctic in boreal spring (Manney et al., 2011) while ozone downward trends in the mid-low latitude lower stratosphere are related to atmospheric dynamics (Chipperfield et al., 2018).

Ozone interacts with radiation and is therefore an important parameter in radiation schemes used in Numerical Weather Prediction (NWP) models (e.g. Hogan et al., 2017), where an improved representation of the ozone field can lead to improvements in weather forecasting or climate simulations. Ozone and the assimilation of ozone retrievals was therefore included in ECMWF's Integrated Forecast System (IFS) system in the late 1990s (Hólm et al., 1999) and a stratospheric ozone parameterization based on Cariolle and Teyssèdre (2007) is still used in the operational ECMWF NWP system where ozone retrievals and ozone sensitive radiances are assimilated (Dethof and Hólm, 2004; Dragani and McNally, 2011; Dragani, 2013). Because this stratospheric ozone parameterisation does not provide realistic tropospheric ozone fields, a comprehensive tropospheric chemistry scheme is used in the CAMS system (Flemming et al., 2015; Flemming et al., 2017).

It is hoped that by adding the assimilation of TROPOMI TCO3 NRT data in the CAMS system, CAMS ozone analyses and forecasts will be improved and that resilience is added in the system against the loss of any of the older instruments whose TCO3 retrievals are currently assimilated by CAMS (see Table 1 below). In a first step, the TROPOMI TCO3 data are monitored passively with the CAMS system. This means they are included in the CAMS data assimilation system, the model fields are interpolated in time and space to the location of the observations and the model equivalents of the observations are calculated allowing temporal and spatial statistics of the differences between the observations and collocated model fields to be determined. However, the data are not actively used in the assimilation at this stage and do not influence the analysis and subsequent forecast yet. We call this 'monitoring' of the observations. In a second step, the active assimilation of the TROPOMI TCO3 data is tested and their impact on the CAMS ozone analysis is assessed by looking at independent validation data.

The differences between the observations and the model fields are called departures. We distinguish between first-guess departures (observations minus model first-guess field) and analysis departures (observations minus analysed field). The first-guess field is a forecast initialised from the previous analysis cycle that is not changed by the analysis increments of the current analysis cycle. If the model fields are stable the departures normally show a relatively smooth behaviour from day to day. Long term monitoring of the departures can disclose errors and biases in the satellite data products, as well as errors or biases in the model. Because the departures are smaller than the absolute observation values, they show up day-to-day changes better than when looking at the absolute model fields or observation values. A sudden jump in the departures on a global scale, which is larger than the instrument noise, can be an indication of problems in the observations or the model.

Including TROPOMI TCO3 data passively in the CAMS system enables us to carry out a continuous quality assessment of the data, to detect biases between different satellite retrievals (e.g. between TCO3 from TROPOMI and the Ozone Monitoring Instrument (OMI) or the Global Ozone Monitoring Experiment-2 (GOME-2)) and allows us to monitor instrument and algorithm stability. The advantage of using an assimilation system to monitor satellite data is that it provides continuous global coverage and allows us to build up global and regional statistics quickly. If the monitoring shows the data to be of good quality, e.g. departures are stable, there are no sudden jumps, the biases with respect to the model are small, assimilation tests with the data usually follow.

This paper is structured in the following way. Section 2 describes the CAMS model and data assimilation system as well as the TROPOMI TCO3 NRT retrievals and how they are used in the CAMS system. Section 3 shows results of monitoring experiments with the TROPOMI TCO3 data, results from first assimilation tests with the data and the validation of the resulting ozone analyses with independent observations. Section 4 gives the conclusions.

## 2 Model and Observations

### 2.1 CAMS model and data assimilation system

The chemical mechanism of the IFS is an extended version of the Carbon Bond Mechanism 5 (CB05, Huijnen et al. 2010) as originally implemented in the Chemical Transport Model (CTM) Transport Model 5 (TM5) and is documented in Flemming et al. (2015) and Flemming et al. (2017). This is a tropospheric chemistry scheme, and for stratospheric ozone the chemical tendencies above the tropopause are computed by a parameterisation based on Cariolle and Teyssèdre (2007). The spatial resolution of the model is approximately 40 km (T511 spectral and 0.35° by 0.35° grid), i.e. coarser than the 3.5 km x 7 km

resolution of the TROPOMI TCO3 data. The CAMS system uses MACCity anthropogenic emissions (Granier et al., 2011), biomass burning emissions from the Global Fire Assimilation System (GFAS, Kaiser et al., 2012) and biogenic emissions form the MEGAN model (Guenther et al., 2006).

ECMWF's IFS uses an incremental 4D-Var data assimilation system going back to Courtier et al. (1994). The data assimilation system for the atmospheric composition fields remains unchanged to the one described in Inness et al. (2015). The atmospheric composition fields are included in the control vector and minimized together with the meteorological control variables. The CAMS NRT system uses 12-hour assimilation windows from 03 UTC to 15 UTC and 15 UTC to 03 UTC and two minimisations at spectral truncations T95 (~ 210 km) and T159 (~ 110 km).

Several ozone retrievals are assimilated in the CAMS NRT system (see Table 1). These include TCO3 retrievals from OMI on the Aura satellite and GOME-2 on Meteorological Operational satellite programme (Metop)-A and -B satellites (referred to as GOME-2AB), $O_3$ profile data from the Microwave Limb Sounder (MLS) and $O_3$ partial columns from Solar Backscatter Ultra-Violet (SBUV/2) and from the Ozone Mapping and Profiler Suite (OMPS). The GOME-2 and OMI TCO3

retrievals are thinned to a horizontal resolution of 0.5° x 0.5° by randomly selecting an observation in a grid box. The MLS profiles and partial column SBUV/2 and OMPS data are used unthinned at present. The $O_3$ retrievals assimilated in the CAMS system are total or partial column data, i.e. integrated layers bounded by a top and a bottom pressure. The model's background column value is calculated as a simple vertical integral between the top and the bottom pressure of the partial or total columns, at the time and location of the observations. Averaging kernels are currently not used for the assimilation of

ozone retrievals in the CAMS system. It is planned to test their use in the future.

A variational bias correction (VarBC) scheme (Dee and Uppala, 2009) where biases are estimated during the analysis by including bias parameters in the control vector was used for several of the $O_3$ retrievals. In this scheme, the bias corrections are continuously adjusted to optimize the consistency with all information used in the analysis. VarBC is applied to the

TCO3 retrievals from OMI and GOME-2 and to the partial column OMPS data, with solar elevation and a global constant as predictors, while the partial column SBUV/2 and the profile MLS data are used to anchor the bias correction, i.e. are assimilated without correction. Experience from past experiments had shown that it is important to have an anchor for the $O_3$ bias correction, to avoid drifts in the $O_3$ fields (Inness et al., 2013).

Variational quality control (Andersson and Järvinen, 1999) and first-guess quality checks are applied to all $O_3$ data in the CAMS system. The variational quality control reduces the weight given to observations in the analysis if they have large background departures. In the first-guess quality check, observations are rejected if the square of the normalized background departure exceeds its expected variance by more than a predefined multiple (5 for most variables).

**2.2 TROPOMI TCO3 NRT retrievals**

TROPOMI has a local overpass time of 13:30 UTC, a ground pixel size of 3.5 km x 7 km for TCO3 and other gases retrieved from the UVVIS, a swath of 2600 km and provides daily global coverage with ~14 orbits per day. For the work in this paper we use NRT TROPOMI TCO3 data (Loyola et al.; 2019b). These include TROPOMI data (V1.0.0) for the period 26 November 2017 to 3 May 2018 V1.0.0 that were reprocessed with the NRT algorithm and NRT TROPOMI data V1.0.0-

V1.1.2 for the period 11 June to 30 November 2018 (see Table 2). No data were acquired at ECMWF from 4 May to 10 June 2018 for technical reasons. The TROPOMI TCO3 retrieval is based on the GDP 4.x algorithm original developed for GOME (van Roozendael et al., 2006), adapted to SCIAMACHY (Lerot et al., 2009) and further improved for GOME-2 (Loyola et

al., 2011; Hao et al., 2014). The major TCO3 algorithm updates for TROPOMI compared to the heritage algorithms used for GOME-2 are the more precise treatment of clouds as scattering layers (Loyola et al., 2018), an optimized wavelength for the calculation of air mass factors (328.2 nm instead of 325.5 nm), better a-priori ozone profile information (including the tropospheric climatology by Ziemke et al. (2011)) and a destriping correction. This destriping correction was introduced because total vertical ozone columns showed small striping structures. The correction factor is based on the ratio between the mean for individual rows and the mean for all rows over a certain region and period. We averaged the total columns within the tropics for January to April 2018 for both all 450 rows individually and over all 450 rows, resulting in an array of 450 numbers, ranging between 0.99 and 1.015. Multiplying the VCD with the correction factor changes the result by about ±1%. The correction factor has been rechecked but no update seemed necessary up to now. The TROPOMI retrieval is described in the S5P/TROPOMI Total Ozone ATBD (Spurr et al., 2018).

We use the following quality checks to remove any outliers of the TROPOMI TCO3 data. Data are only used if:

1. Value of quality flag given in the data ('qa flag') between [0, 100]
2. Ozone values between [0,900 DU]
3. Surface altitude between [-399, 8850m]
4. Cloud Fraction between [0,1]
5. Latitude between
   a. [-90°, 90°] for the monitoring assimilation (section 3.1)
   b. [-60°, 60°] in the active assimilation (section 3.2)

Data that pass the above four checks are flagged as 'good' and used for the studies presented in this paper. In the current TROPOMI TCO3 products V1.1.x (Pedergnana et al., 2018) the 'qa flag' will allow the user to identify good quality data, but this was not yet the case in V1.0.0. For consistency with the earlier period the qa value filter was not applied for version 1.1.2 data in this study.

Because the horizontal resolution data of TROPOMI (3.5 km x 7 km) is higher than the model resolution of T511 (about 40 km x 40 km) the TROPOMI data are not spatially representative for the model grid boxes. To overcome this representativeness error, the data are converted to so called 'super-observations' before they are included in the CAMS system. For this super-obbing the data are averaged to the T511 resolution of the model. The averaging is carried out for all 'good' data and the errors of the data are averaged in the same way as the observations. The super-obbing reduces the random errors in the data. In the past a 'random' thinning to 0.5°x0.5° was used in the CAMS system and this is still applied to the TCO3 retrievals from GOME-2 and OMI. The super-obbing applied to TROPOMI data has the advantage that it does not simply throw out the majority of the observations but uses the information from all good data to create average observations. In future, super-obbing will also be tested for the other ozone datasets. An example of TROPOMI TCO3 NRT data at full resolution and super-obbed to T511 is shown in Figure 1. The super-obbing reduces the number of 'good' TROPOMI TCO3 data from about 15-16 million per day to about 500,000 while still making use of the information given by all good data.

## 3 Results

Two experiments were run with NRT TROPOMI TCO3 data super-obbed to T511 horizontal resolution. In the first control experiment (CTRL, section 3.1) the TROPOMI data were included passively, in the second one (ASSIM, section 3.2) they

were actively assimilated. The experiments cover the period from 26 November 2017 to 30 November 2018 and were run using CY45R1 of the CAMS system. We look at timeseries for the whole period and also at fields averaged over the periods February to April (FMA) and September to November (SON) 2018.

## 3.1 Monitoring of TROPOMI TCO3 NRT data

Figure 2a shows a timeseries of zonal mean weekly averaged TCO3 values from TROPOMI for the period 26 November 2017 to 30 November 2018. The timeseries shows a realistic evolution of the ozone field with high column values in the NH during winter and spring, low values in the Tropics throughout the year, higher values in the circum-Antarctic band and the lowest absolute values over the Antarctic during the ozone hole from August to November 2018. We also see a longer period without TROPOMI data in January 2018 when the instrument was undergoing calibration activities. From mid-February

2018 onwards (outside the data gap in May/ June when data were not acquired by ECMWF) the number of data is more stable (not shown) except from one week with low data numbers in March. To assess the quality of the TROPOMI data they are compared with the other three TCO3 retrievals that are routinely assimilated in the CAMS NRT system, i.e. OMI and GOME-2AB in Figure 2b-2d. The TROPOMI NRT DOAS retrieval is a further development of the operational AC-SAF GOME-2AB DOAS, therefore a better agreement can be expected for the comparison for those sensors compared to OMI

where a different algorithm (TOMS-like) is applied. The differences between TROPOMI and OMI are positive (i.e. TROPOMI values higher than OMI) in most latitude bands throughout the timeseries. Negative differences (i.e. TROPOMI values lower than OMI) are found at the northern ends of the orbits from December 2017 to April 2018 and September to November 2018, and also south of 60ºS, particularly from March to October when UVVIS retrievals generally have larger problems because of the illumination conditions and the icy surfaces.  On top of that, the TROPOMI retrievals have a larger

bias at high latitudes because the current NRT algorithm uses a surface climatology that does not fully represent the actual snow/ice conditions. Over large parts of the timeseries the differences are below 4-6 DU corresponding to less than 2%. Larger departures are seen during some of the commissioning and differences of up to 20 DU are seen north of 40ºN during the first half of the timeseries. However, TCO3 values are also larger at this time (Figure 2a) so this still corresponds to agreements within about 5%. The differences between TROPOMI and GOME-2AB (Figure 2c and d) are smaller than the

differences with OMI between 60ºN and 60ºS, but larger negative differences are found at the northern end of the orbits, especially south of 60ºS. Again, larger departures are seen during some of the commissioning phase. Apart from these periods the differences between the TROPOMI and GOME-2AB data for latitudes between 60ºN and 60ºS are between 2-6 DU or less than about 2%.

Figure 3 shows the mean TROPOMI TCO3 fields averaged for FMA and SON 2018 and the differences between TROPOMI, OMI and GOME-2AB. TROPOMI shows high TCO3 values in FMA during spring in the NH and low values over the Antarctic ozone hole during SON. The Figure shows that there are large differences between TROPOMI and GOME-2AB polewards of about 60º which seem to be mainly negative over ice and sea and positive over land. During FMA there are also large differences in the NH north of about 40ºN. However, these differences are still within about 20 DU (or

less than 5%) in most areas south of 60ºN for GOME-2B and only slightly larger for GOME-2A. These differences come mainly from the surface albedo climatology that is used in the TROPOMI NRT retrieval algorithm of the V1.0.0 - V1.1.2 data. The employed surface albedo climatology, based on OMI data (Kleipool et al., 2008), has a spatial resolution of 0.5° x 0.5° which seems coarser than the spatial resolution of the TROPOMI pixels. Consequently, surface albedo structures are found in the obtained TCO3 results, particularly over the polar regions where the surface albedo climatology sometimes has

very few grid cells marked as no snow or ice (reflectivity 0.05) whereas the reflectivity is close to one for the neighbouring ones with snow. In the future, it is planned to replace this coarse climatology with a new surface albedo retrieval using S5P data (Loyola et al., 2019a). For all three instruments, the differences in the NH are smaller in SON than FMA. During

summer, when there is no snow cover, the resolution of the surface albedo is less of an issue and larger positive differences between TROPOMI and GOME-2AB are confined to Alaska and Kamchatka. We expect the largest problems to occur in spring and autumn when the snow cover changes locally. Equatorward of those areas the differences between the instruments are smaller. Over oceans TROPOMI and GOME-2AB mainly agree to within ± 4 DU, i.e. less than about 2% of the tropical TCO3 values, with slightly larger differences over land (up to 10 DU, ~ 5%). During SON, negative differences of up to -6 DU are found over the maritime continent. This is probably caused by the update of the a priori in the TROPOMI algorithm V1.1.2, as the new one takes the tropospheric ozone wave one structure into account. The differences between TROPOMI and OMI are slightly larger than the differences between GOME-2AB and TROPOMI, but still less than about 8 DU (or 4%) over larger parts of the Globe, with TROPOMI generally higher than OMI except over the Maritime continent. Over Antarctica and some areas north of 40ºN positive and negative deviations are found. Because the differences over Antarctica show similar structures for OMI and GOME-2AB they are likely to point to problems with the TROPOMI retrievals using the OMI surface climatology rather than the other datasets.

Figure 4 shows maps of the standard deviation of the four TCO3 retrievals over the whole period from 26 November 2017 to 3 November 2018. All retrievals show the same features with highest variability in the high northern latitudes and over Antarctica where TCO3 values vary most during the course of a year and the lowest variability in the Tropics.

We now look at differences between TROPOMI and the CAMS ozone analysis, i.e. analysis departures. These departure plots show problems in the TROPOMI data more clearly than the comparison between instruments in Figure 3, because they are not affected by issues from two different retrievals. Nevertheless, the main findings from Figure 5 are similar to those from Figure 3. Figure 5a shows a timeseries of zonal mean weekly averaged TROPOMI analysis departures for the period 26 November 2017 to 30 November 2018 and like Figure 3 has the largest, mainly negative, departures polewards of 60º. Between 50ºN and 60ºS the zonal mean departures are within ±2-4 DU during most of the timeseries, i.e. less than 2% of the zonal mean TCO3 (Figure 2a). Larger positive departures are found between 50-60ºN until August 2018 and between 50-60ºS after the end of March. After the latest algorithm change to V1.1.2 in August, the zonal mean departures are less than ±2 DU between 60ºN and 60ºS most of the time and also smaller than -4 DU for most of the time between 60-90ºS. During FMA (Figure 5b) TROPOMI is lower than the CAMS analysis south of 60ºS and over land or snow/ice north of 60ºN. TROPOMI is considerably higher than the CAMS analysis over land in the NH north of about 40ºN with differences between 20-40 DU in places, a result of the issues with the surface albedo climatology discussed above. However, as TCO3 values are also large in these areas (see Figure 3a1), this is still within about 10% of the mean observation values. Large positive TROPOMI TCO3 departures are also seen over the Himalayas in FMA. In the other areas, TROPOMI agrees better with the model, with positive departures over the tropical Atlantic, Africa and South America (up to 8 DU, about 4%) and small negative departures elsewhere. During SON (Figure 5c) the departures are generally smaller than during FMA, especially over land in the NH. The largest negative departures are found over sea/ice north of 60ºN (up to -20 DU). Over the Maritime continent larger negative departures (up to -8 DU) are found during SON and FMA. As mentioned above this is probably caused by the update of the a priori in the TROPOMI retrievals that takes the tropospheric ozone wave one structure into account. We will show in section 3.2.2 that the assimilation of TROPOMI improves the CAMS results compared to tropospheric ozone sondes and IAGOS measurements in the Tropics, suggesting these differences point to a model bias rather than a problem with the data.

Table 3 lists mean biases and their standard deviations for the first half (November 2017-May 2018) and the second half (June - November 2018) of the timeseries. Averaged over November 2017-May 2018 TROPOMI shows a mean bias with respect to CAMS of -1.07±17.3 DU between 90-60ºN, 2.10±9.47 DU between 60-30ºN, 0.06±3.83 DU between 30ºN-30ºS, -

0.05±4.95 DU between 30-60ºS and -6.81±7.32 DU between 60-90ºS. The mean bias between 90-60ºN is relatively small because the positive biases over land and the negative ones over ice compensate (see Figure 5a and b). This is also illustrated by the large standard deviation between 90-60ºN. Table 3 also lists the values from the other three TCO3 retrievals and shows that the mean biases of TROPOMI are larger than GOME-2AB's between 90-30ºN and between 60-90ºS, but smaller

between 30ºN and 60ºS. TROPOMI has smaller mean biases than OMI in all areas except 60-90ºS. The standard deviations of the TROPOMI departures are larger than GOME-2AB's in all areas and larger than OMI's between 90-30ºN and between 60-90ºS but smaller than OMI's between 30ºN and 60ºS. It should be born in mind though, that 'used' data are shown for GOME-2AB and OMI and the CAMS analysis draws to the data thus reducing the standard deviation of the departures, while 'good' data are shown for TROPOMI and the TROPOMI data no not affect the CAMS analysis. For June to

November 2018 TROPOMI has mean bias of -1.46±10.40 DU between 90-60ºN, 0.31±6.00 DU between 60-30ºN, -0.47±3.88 DU between 30ºN-30ºS, 0.82±10.10 DU between 30-60ºS and -2.39±6.99 DU between 60-90ºS. Again, the TROPOMI biases and standard deviations of the departures are larger than GOME-2AB's in most areas while the biases are smaller than OMI's between 90ºN-30ºS and larger between 30º-90ºS.

Figure 6 shows scatter plots of TROPOMI TCO3 analysis departures for the period 26 November 2017 to 30 November 2018 against latitude, solar elevation and scan position. Such plots can be very useful in identifying retrieval problems depending on the chosen parameters. In FMA, the scatter plot against latitude shows small mean departures between about 60ºS and 45ºN, positive departures between 45ºN and 70ºN and negative departures polewards of 70ºN and 60ºS. This agrees with the averaged analysis departures shown in Fig. 5b and the Hovmoeller plot in Fig. 5a and illustrates that there is a

problem with the retrievals at high latitudes. The plot also shows that there is a large scatter polewards of 50ºN and 60ºS. Larger scatter at high latitudes is also seen for OMI and GOME-2AB if all 'good' data are plotted (not shown). In SON, the mean departures plotted against latitude are generally small and do not show high values between 45-70ºN any more. They are still negative polewards of 70ºN and 60ºS, but smaller than in FMA. In FMA, the TROPOMI departures show a strong dependency on solar elevation with increasingly negative departures at solar elevations below 25º. The SOE dependency is

smaller in SON. For a sun synchronous orbiting satellite, the SOE is mainly a function of the latitude, therefore this deviation might be caused by the latitudinal deviation discussed above. Departures in both seasons vary slightly depending on the scan position with increasingly negative departures towards the edges of the scan, especially on the western side of the scan. There is no dependency of the departures on cloud cover or cloud top pressure (not shown).

**3.2 Assimilation tests with TROPOMI TCO3 NRT data**

We showed in section 3.1 that the TROPOMI TCO3 data are of good quality over large parts of the globe, but that there are some issues at high latitudes and low solar elevations, especially in FMA. The biases we observe outside those regions are of similar magnitude to the biases of the other total column data sets assimilated in CAMS (see Table 3) and we therefore do not expect any problems with the assimilation of TROPOMI NRT TCO3 if we bias correct the data and blacklist them

appropriately. Hence, assimilation tests are carried out with the TROPOMI NRT TCO3 data for the period 26 November 2017 to 30 Nov 2018, blacklisting them for solar elevations less than 10º and poleward of 60º. Restricting the assimilated data between 60°S and 60°N excludes the "ozone –hole" observation in these tests. Variational bias correction is applied to the data in the same way as it is used for the other TCO3 data, i.e. with solar elevation and a global constant as predictors. The choice of these bias correction parameters can be altered in the future if needed.

### 3.2.1 Impact of the TROPOMI assimilation

Figure 7 shows timeseries of global mean weekly averaged TROPOMI, OMI and GOME-2AB TCO3 departures, bias correction, standard deviation of departures and number of observations between 26 November 2017 and 30 November 2018 for 'used data', i.e. the data that fulfil the blacklist criteria and quality checks listed in Table 1 and pass the variational

quality control and first-guess checks applied by the IFS (see section 2.1). The figure shows that the TROPOMI bias correction successfully removes the biases between the data and the model, so that the bias corrected analysis departures are small. The bias correction calculates maximum values of about 1 DU in the global mean with the largest positive values between June and August and the largest negative values in November 2018. The magnitude of the global mean bias correction that is applied to TROPOMI is smaller than that of the other three TCO3 retrievals. Figure 7 shows that the

analysis is drawing to the TROPOMI data (and the other three datasets), i.e. analysis departures are smaller than the first-guess departures and the standard deviation of the departures is reduced. About 2.4 million TROPOMI observations are used every week which is 10x as many observations as from OMI, 5x as many as from GOME-2A and 3x as many as from GOME-2B.

Table 4 lists mean biases and standard deviations from ASSIM for all four TCO3 retrievals for the period 26 November 2017 to 3 May 2018 and 11 June to 30 November 2018 for 'used' data. It shows that TROPOMI TCO3 bias and standard deviation values are reduced in all three areas compared to the values in CTRL (see Table 3) as the analysis is drawing to the data. The fit to the other datasets is slightly degraded in some areas and improved in others when TROPOMI data are also assimilated (see Table 4), but overall the differences between the biases and the standard deviations of the departures of

GOME-2AB and OMI from ASSIM and CTRL are small.

Figure 8a shows a timeseries of the zonal mean weekly averaged bias correction that is applied to TROPOMI data for the period 26 November 2017 to 30 November 2018. The Figure illustrates how the bias correction changes with time as it adapts to the data and that the mean bias-corrected TROPOMI analysis departures for FMA (Figure 8b) and SON (Figure 8c)

are small compared to CTRL (Figure 5b and c) as the analysis is drawing to the TROPOMI data. Some larger positive departures remain over land in the NH in FMA where observation outliers are given less weight by the analysis.

Figure 9 shows the mean TCO3 fields from ASSIM for FMA and SON 2018 as well as the absolute and relative differences between ASSIM and CTRL. It illustrates that the impact of the TROPOMI assimilation in relative terms is small with

relative differences of less than 2% everywhere and less than 1% in most areas. The absolute differences are largest over land in the NH in FMA with ASSIM up to 10 DU higher than CTRL. However, the absolute TCO3 values are also largest then. Positive differences are also found in an area stretching from South America over the Atlantic to Africa in FMA and SON and in small bands around 60°S. In most other areas, the differences are below -2 DU and negative.

Figure 10 shows cross sections of zonal mean relative $O_3$ mixing ratio differences from ASSIM minus CTRL averaged over FMA and SON. Again, the impact of TROPOMI assimilation is small with the largest relative differences found in the troposphere. Here the TROPOMI data act to lower the ozone values in ASSIM in the zonal mean. In FMA the impact is less than 1% everywhere. In SON, the differences in the troposphere are slightly larger and reach values of up to -3% near the surface in NH midlatitudes and over the South Pole. Note that no TROPOMI data were assimilated south of 60°S so the

changes seen here come from transport. Also, note that the absolute $O_3$ values in the lower troposphere over the Antarctic are small.

### 3.2.2 Validation with independent observations

To assess if the assimilation of TROPOMI TCO3 retrievals improves or degrades the CAMS analysis, the $O_3$ fields from ASSIM and CTRL are compared with independent observations. We use for comparison the following datasets. (1) Brewer spectrometer measurements obtained from the World Ozone and Ultraviolet Radiation Data Centre (WOUDC). The Brewer data are well calibrated with a precision of 1% (Basher, 1982). (2) Ozone sonde data from a variety of data centres: WOUDC, Southern Hemisphere ADditional OZonesondes (SHADOZ), Network for the Detection of Atmospheric Composition Change (NDACC), and campaigns for the Determination of Stratospheric Polar Ozone Losses (MATCH). The precision of electrochemical concentration cell (ECC) ozone sondes is on the order of ±5% in the range between 200 and 10 hPa, between −14% and +6% above 10 hPa, and between −7% and +17% below 200 hPa (Komhyr et al., 1995). Larger errors are found in the presence of steep gradients and where the ozone amount is low. The same order of precision was found by Steinbrecht et al. (1998) for Brewer–Mast sondes. (3) Ozone profiles from instruments mounted on commercial aircraft from the In-service Aircraft for a Global Observing System (IAGOS). The IAGOS ozone data have a detection limit of 2 ppbv and a precision of ± (2 ppbv + 2 %) (Marenco et al.,1998). (4) Ground-based data from the World Meteorological Organisation's Global Atmosphere Watch (GAW) surface observation network (e.g., Oltmans and Levy, 1994; Novelli and Masarie, 2014). The GAW observations represent the global background away from the main polluted areas. GAW $O_3$ data have a precision of ±1 ppbv (Novelli and Masarie; 2014).

Figure 11 shows timeseries of the weekly averaged TCO3 biases from ASSIM and CTRL against Brewer measurements averaged over the Globe and NH midlatitudes for the period 26 November 2017 to 30 November 2018. The Figure shows a generally good agreement of both experiments with the Brewer data with maximum biases of less than 6 DU. It confirms that the impact of the TROPOMI assimilation in the CAMS system is small with differences between ASSIM and CTRL of less than 1 DU in the total column. Despite being small, the impact usually leads to an improved fit to the WOUDC data in ASSIM.

Compared with ozone sondes averaged over FMA (Figure 12) and SON (Figure 13) the impact in relative terms is also small. However, an improved fit to the data is seen in ASSIM in the Tropics during SON when the positive bias seen in CTRL is reduced. Ozone profiles from ASSIM and CTRL are also compared with IAGOS aircraft data (Figure 14). Because not many IAGOS profiles were available during October and November 2018 we show FMA and July-September (JAS) 2018. In both seasons, we see a positive impact from the assimilation of TROPOMI TCO3 data over West African airports where the negative bias seen in CTRL is reduced when assimilating TROPOMI TCO3 data. This increase in tropospheric $O_3$ agrees with the increased TCO3 seen in ASSIM over Africa in FMA and SON (Figure 9), but does not show up in the zonal mean cross sections (Figure 10).

Finally, we compare ASSIM and CTRL with GAW $O_3$ surface observations over Europe in Figure 15 and again see only a small, slightly negative impact. However, only very few GAW stations were available and the mean are calculated from between 3-5 stations over Europe and more data would be needed for a meaningful validation of surface $O_3$.

On the whole, the impact of the TROPOMI assimilation in the CAMS system is relatively small because the CAMS analysis is already well constrained by the other $O_3$ data sets that are assimilated routinely which are a combination of TCO3 data (OMI, GOME-2AB), $O_3$ layers (SBUV/2, OMPS) and $O_3$ profiles (MLS) (see Table 1). If no other $O_3$ data were available and only TROPOMI TCO3 data were assimilated the impact on the CAMS $O_3$ analysis would be larger. To confirm that

TROPOMI could serve as a good replacement if one of the older TCO3 instruments (OMI, GOME-2AB) failed two further experiments were run for the period 26 November 2017 to 30 April 2018: one mimicking the configuration of CTRL, but without OMI (CTRL-OMI) and the other mimicking the configuration of ASSIM without OMI (ASSIM-OMI). Compared to ozonesondes and IAGOS data the differences between these experiments and ASSIM and CTRL are very small indeed. The largest differences between the four experiments are found in the SH midlatitudes when compared with ozone sondes (Figure 16a) and over West African airports compared with IAGOS (Figure 16b). Even here, the differences between ASSIM and ASSIM-OMI are small and both fit the independent observations better than CTRL and CTRL-OMI. There is even a sign that removing OMI leads to a small improvement in the fit to IAGOS over West Africa. In all other areas the differences between the experiments with and without OMI were negligible when compared to sondes or IAGOS. These findings agree with results from longer observation system experiments that were carried out with the CAMS system for the years 2013 and 2014 in a different context (not shown) which showed only small changes to the CAMS $O_3$ analysis if one of the TCO3 instruments was removed confirming that the CAMS analysis is well constrained and that there is some redundancy in the system. We are therefore confident that TROPOMI will be able to counterbalance the loss of one of the older TCO3 instruments. Removing MLS $O_3$ profiles has a much larger (negative) impact on the CAMS $O_3$ analysis (e.g. Flemming et al., 2011) and TROPOMI would not be able to replace the MLS profiles as it does not provided data with similar vertical resolution.

The main advantage of assimilating TROPOMI into the CAMS system seems to be the improvement of tropospheric ozone. This is because MLS defines the stratosphere whereas OMI, GOME-2 and TROPOMI are also sensitive to the troposphere and add extra information here (see also Lefever et al., 2015). Adding TROPOMI to the CAMS system fits the CAMS analysis better to independent tropospheric data.

**4 Conclusions**

TROPOMI NRT TCO3 retrievals for the period 26 November 2017 to 30 November 2018 have been included in the CAMS data assimilation system to first monitor the data and to then carry out assimilation tests with them. The TROPOMI data used for the work presented in this paper were TROPOMI TCO3 data (V1.0.0) that had been reprocessed with the NRT algorithm until 3 May 2018 and NRT TROPOMI (V1.0.0 to v1.1.2) data for the period 11 June to 30 November 2018 (Loyola et al., 2019b). TROPOMI was still in its commissioning phase until 24 April 2018, but even the early TROPOMI TCO3 data generally agreed well with the CAMS analysis over large parts of the Globe and were of good enough quality to test their use in the CAMS system.

Monitoring of TROPOMI TCO3 data in the CAMS system has shown that the data are of good quality over large parts of the Globe. The TROPOMI TCO3 biases relative to the CAMS $O_3$ analysis are of similar magnitude to biases of OMI and GOME-2AB TCO3 between 60ºN and 60ºS and TROPOMI agrees with the CAMS ozone analysis to within 2% over large parts of the Globe in weekly mean zonal mean timeseries and averaged over FMA and SON 2018. However, there are problems with the TROPOMI TCO3 NRT retrievals at high latitudes, at low solar elevations and over snow/ice (e.g. Antarctica or ice-covered areas in NH). These differences, which are most prominent over land in the NH north of 45ºN before May 2018, come mainly from the surface albedo climatology that is used in the TROPOMI NRT retrieval algorithm and has a spatial resolution of 0.5° x 0.5° which is coarser than the spatial resolution of the TROPOMI pixels. It is planned to replace this climatology with a climatology based on TROPOMI data when data for a long enough period are available. The bias of TROPOMI TCO3 has a dependency on solar elevation, with increasingly negative biases at solar elevations less than 25º, especially in FMA 2018. During SON 2018 the dependency on solar elevation is considerably smaller. The bias of

TROPOMI TCO3 relative to CAMS also depends slightly on scan position, with increasingly negative bias towards the western edge of the scan.

Relative to CAMS and averaged over the period 26 November 2017 to 3 May 2018, TROPOMI TCO3 NRT data show a mean bias with respect to CAMS of $-1.07\pm17.3$ DU between 90-60ºN, $2.10\pm9.47$ DU between 60-30ºN, $0.06\pm3.83$ DU between 30ºN-30ºS, $-0.05\pm4.95$ DU between 30-60ºS and $-6.81\pm7.32$ DU between 60-90ºS. For June to November 2018 TROPOMI has mean bias of $-1.46\pm10.40$ DU between 90-60ºN, $0.31\pm6.00$ DU between 60-30ºN, $-0.47\pm3.88$ DU between 30ºN-30ºS, $0.82\pm10.10$ DU between 30-60ºS and $-2.39\pm6.99$ DU between 60-90ºS. This paper illustrates the power of using a global assimilation system to monitor new satellite products, as it provides continuous global coverage, allows us to build up global and regional statistics quickly and can help to identify problems with the retrievals (e.g. biases against solar elevation, latitude, scan position, surface albedo dependencies, etc.) that might be more difficult to discover when comparing TROPOMI retrievals against sparse in-situ observations.

Assimilation tests were carried out with the TROPOMI TCO3 data, blacklisting them poleward of 60º and at solar elevations less than 10º, and applying ECMWF's variational bias correction scheme to the data with solar elevation and a global constant as predictors. These assimilation tests showed that the bias correction successfully removed the biases between the model and the data. Overall, the impact of the TROPOMI data in the CAMS assimilation system was found to be relatively small, because the ozone analysis is already well constrained by several other ozone data sets that are assimilated routinely (OMI, GOME-2AB, MLS, SBUV/2, OMPS). Mean differences between a run with and without assimilation of TROPOMI TCO3 NRT data over the FMA and SON 2018 were less than 2% for TCO3 everywhere and less than 1% in most areas. For average zonal mean $O_3$ mixing ratio profiles the differences between ASSIM and CTRL were less than 3% with the largest relative differences found in the troposphere where the assimilation of TROPOMI TCO3 data led to decreased ozone values in the zonal mean. Zonal mean differences in the stratosphere where less than 1%.

ASSIM and CTRL show only small differences when compared with independent ozone observations, however these differences are mainly positive. There is a slightly improved fit to WOUDC Brewer TCO3 data in ASSIM. The largest impact of the TROPOMI assimilation was found over West African airports, where the assimilation led to increased ozone values in the troposphere and a reduced negative bias against IAGOS aircraft profiles in FMA and SON 2018. A positive impact in the tropical troposphere was also seen against ozone sondes in SON 2018 where the zonal mean positive bias was reduced. It seems the main advantage of assimilating TROPOMI into the CAMS system is the improvement of tropospheric ozone. This is because MLS defines the stratosphere whereas OMI, GOME-2AB and TROPOMI are also sensitive to the troposphere and add extra information here (see also Lefever et al., 2015). Adding TROPOMI to the CAMS system improves the fit of the CAMS analysis to independent tropospheric data and makes the CAMS system more resilient against the loss of any of the older TCO3 instruments that are currently assimilated. Assimilation tests show that good results are achieved when replacing OMI with TROPOMI.

Due to the limitations of the current TROPOMI TCO3 NRT product that uses a OMI climatology for the surface properties, ozone data had to be blacklisted at high latitudes in this study. Future algorithm updates dealing with a better treatment of the surface albedo (Loyola et al., 2019a) will improve the retrieval quality at high latitudes and should allow the data to be used up to the poles. The V1.1.2 data used after 8 August 2018 already show smaller departures south of 60ºS. Note that the TROPOMI TCO3 offline algorithm does not have the limitation due to the surface albedo climatology seen in the NRT product because the surface albedo is fitted as part of the retrieval.

TROPOMI TCO3 NRT data were included passively in the operational CAMS system on 13 July 2018, the day the data were officially released by ESA, and have been monitored routinely by CAMS ever since (see https://atmosphere.copernicus.eu/charts/cams_monitoring). Because of the small, but positive impact of the TROPOMI TCO3 assimilation on the CAMS ozone analysis shown in this paper it was decided to actively include the TROPOMI TCO3 NRT data in the operational NRT CAMS analysis, and the routine assimilation of the data in the operational CAMS analysis began on 4 December 2018.

**Author Contributions**

A. Inness carried out the experiments described in the paper, the validation of the resulting analysis fields and wrote the manuscript, R. Ribas set up the S5P processing chain at ECMWF which included coding and testing the BUFR converter needed to ingest the TROPOMI TCO3 data in the ECMWF data system, J. Flemming helped with the development of the IFS chemistry module, D. Loyola, W. Zimmer, K.-P. Heue, J. Xu, P. Valks, C. Lerot and M. van Roozendael developed the TROPOMI TCO3 retrieval algorithm and the operational processing chain at DLR. All co-authors gave useful comments during the writing of the paper.

**Acknowledgements**

Thanks to the DLR colleagues Fabian Romahn and Mattia Pedergnana working on the operational UPAS system for generating TROPOMI TCO3 products and thanks to Maximilian Schwinger and the PDGS team at DLR responsible for the Sentinel-5 Precursor payload data ground segment. Thanks to Luke Jones for help with the plotting of ozone sondes, GAW and IAGOS data. Thanks to the data providers of the data assimilated in the CAMS reanalysis and the data used for the validation studies in this paper. The TROPOMI total ozone data are generated at DLR on behalf of EU/ESA. The GOME-2 total ozone data assimilated in CAMS are provided by DLR in the framework of the EUMETSAT AC-SAF project. The Copernicus Atmosphere Monitoring Service is operated by the European Centre for Medium-Range Weather Forecasts on behalf of the European Commission as part of the Copernicus programme (http://copernicus.eu).

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

| Instrument/ Satellite | Data product | Data provider/version | Blacklist criteria / thinning | VarBC predictors | Reference |
|---|---|---|---|---|---|
| GOME-2/ Metop-A | TCO3 | AC-SAF/ DLR GDP4.8 | QF>0 SOE<6º Thinned to 0.5ºx0.5º | Solar elevation Global constant | Hao et al. (2014), Valks et al. (2017) |
| GOME-2/ Metop-B | TCO3 | AC-SAF/ DLR GDP4.8 | QF>0 SOE<6º Thinned to 0.5ºx0.5º | Solar elevation Global constant | Hao et al. (2014), Valks et al. (2017) |
| MLS/ Aura | $O_3$ profiles | NASA V3.4 | QF>0 No thinning | Not applied | Schwartz et al. (2015) |
| OMI/ Aura | TCO3 | NASA V883 | QF>0 SOE<10º Thinned to 0.5ºx0.5º | Solar elevation Global constant | Liu et al. (2010) |
| OMPS (nadir)/ Suomi NNP | $O_3$ partial columns | NOAA/ Eumetsat V1r0 | QF>0 SOE<10º No thinning | Solar elevation Global constant | Flynn et al. (2014) |
| SBUV/2/ NOAA-19 | $O_3$ partial columns | NOAA V8 | QF>0 SOE<6º No thinning | Not applied | Bhartia et al. (1996), McPeters et al. (2013) |
| TROPOMI/ Sentinel-5P | TCO3 | ESA/ DLR V1.0.0-V1.1.2 (see Table 2) | QF>0 SOE<10º Abs(LAT)<60º Super-obbed to T511 | Solar elevation Global constant | Loyola et al. (2019 a) |

**Table 1: O₃ satellite retrievals used in this paper. QF= quality flag given by data providers, SOE= Solar Elevation, LAT: Latitude, VarBC: Variational bias correction. The blacklist criteria describe when data were not used.**

| Period | Version number | Algorithm | Description of changes |
|---|---|---|---|
| 20171126-20180503 | V1.0.0 | Reprocessed with NRT algorithm | N/A (original algorithm) |
| 20180611-20180718 | V1.0.0 | NRT | N/A (original algorithm) |
| 20180718-20180808 | V1.1.1 | NRT | Minor bugfixes, no algorithm changes. QA_values introduced |
| 20180808-20181130 | V1.1.2 | NRT | Bug fix to time variable |

**Table 2: Version numbers of TROPOMI data used in this study.**

| Instrument | Period | 90º-60ºN | 60º-30ºN | 30ºN-30ºS | 30º-60ºS | 60º-90ºS |
|---|---|---|---|---|---|---|
| TROPOMI | Nov-May | -1.07± 17.30 | 2.10±9.47 | 0.06±3.83 | -0.05±4.95 | -6.81±7.32 |
| OMI | Nov-May | -2.85±8.11 | -2.70±7.99 | 0.18±7.45 | 0.83±7.35 | 2.17±6.42 |
| GOME-2A | Nov-May | 0.81±6.35 | -0.60±5.88 | 0.06±3.14 | -0.10±3.40 | 0.86±3.38 |
| GOME-2B | Nov-May | 0.29±6.25 | 0.42±6.14 | 0.16±2.97 | -0.19±3.33 | -0.46±3.31 |
| TROPOMI | Jun-Nov | -1.46±10.40 | 0.31±6.00 | -0.47±3.88 | 0.82±10.10 | -2.39±6.99 |
| OMI | Jun-Nov | -1.57±7.22 | -1.63±7.77 | 0.48±7.15 | 0.73±7.95 | 2.07±6.65 |
| GOME-2A | Jun-Nov | 0.19±5.11 | -0.25±4.51 | 0.02±3.55 | 0.24±5.08 | 0.67±3.56 |
| GOME-2B | Jun-Nov | 0.68±4.80 | 0.53±4.45 | -0.18±3.27 | 0.04±5.08 | -0.29±3.65 |

**Table 3: Mean bias and standard deviations of the TCO3 retrievals against the CAMS ozone analysis in DU from the control experiment (CTRL) for the periods 26 November 2017 to 3 May 2018 and 11 June to 30 November 2018. Green numbers mark**

**where the biases and standard deviations of the other TCO3 datasets are smaller than TROPOMI's, red marks where they are larger. Shown are 'good' data for TROPOMI and 'used' data for the other instruments.**

| Instrument (used data) | Period | 90º-60ºN | 60º-30ºN | 30ºN-30ºS | 30º-60ºS | 60º-90ºS |
|---|---|---|---|---|---|---|
| TROPOMI | Nov-May | Not used | 0.51±6.64 | 0.07±2.44 | 0.003±3.24 | Not used |
| OMI | Nov-May | -3.19±8.09 | -3.34±8.25 | 0.22±7.42 | 1.08±7.21 | 2.20±6.41 |
| GOME-2A | Nov-May | 0.71±6.38 | -0.90±5.88 | 0.06±3.13 | 0.02±3.56 | 0.86±3.38 |
| GOME-2B | Nov-May | 0.20±6.28 | 0.14±6.27 | 0.20±3.06 | -0.13±3.43 | -0.47±3.31 |
| TROPOMI | Jun-Nov | Not used | 0.07±4.09 | 0.03±2.64 | 0.29±4.63 | Not used |
| OMI | Jun-Nov | -1.76±7.22 | -1.82±7.87 | 0.67±7.10 | 0.25±7.75 | 2.03±6.63 |
| GOME-2A | Jun-Nov | 0.17±5.12 | -0.15±4.60 | 0.05±3.77 | 0.03±5.24 | 0.59±3.61 |
| GOME-2B | Jun-Nov | 0.71±4.80 | 0.68±4.67 | -0.16±3.45 | -0.29±5.16 | -0.41±3.69 |

5    **Table 4: Mean bias and standard deviations of the TCO3 retrievals against the CAMS ozone analysis in DU from the assimilation experiment (ASSIM) for the periods 26 November 2017 to 3 May 2018 and 11 June to 30 November 2018 for 'used' data. Green numbers mark where the biases or standard deviations are smaller in ASSIM than in CTRL (Table 3), red numbers mark where they are larger.**

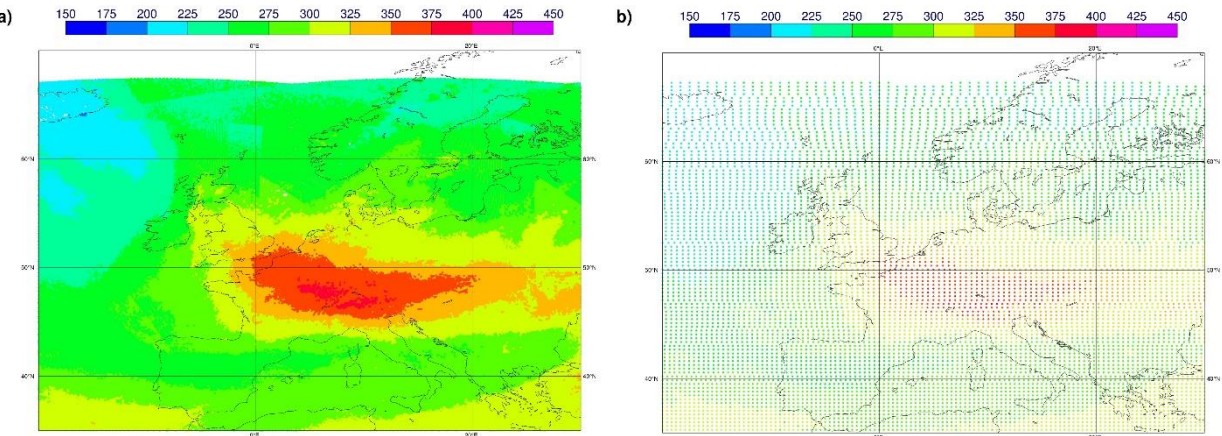

**Figure 1: TROPOMI NRT TCO3 in Dobson Units (DU) at (a) full resolution and (b) super-obbed to the model resolution of T511 on 20171129, 12z over Europe.**

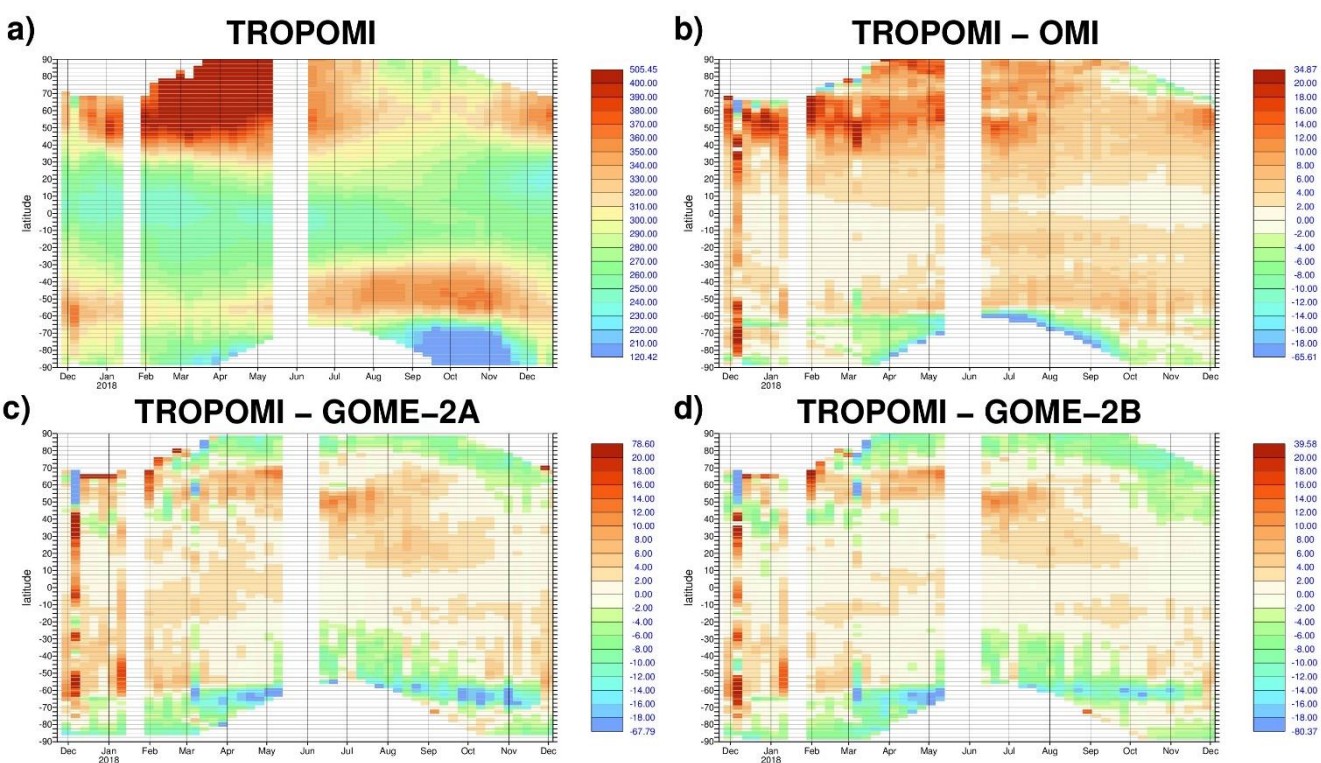

 **Figure 2: (a) Timeseries of zonal mean weekly averaged TROPOMI NRT TCO3 for the period 26 November 2017 to 30 November 2018 and differences between (b) TROPOMI and OMI, (c) TROPOMI and GOME-2A and (d) TROPOMI and GOME-2B TCO3. Shown are 'good data. All values in DU.**

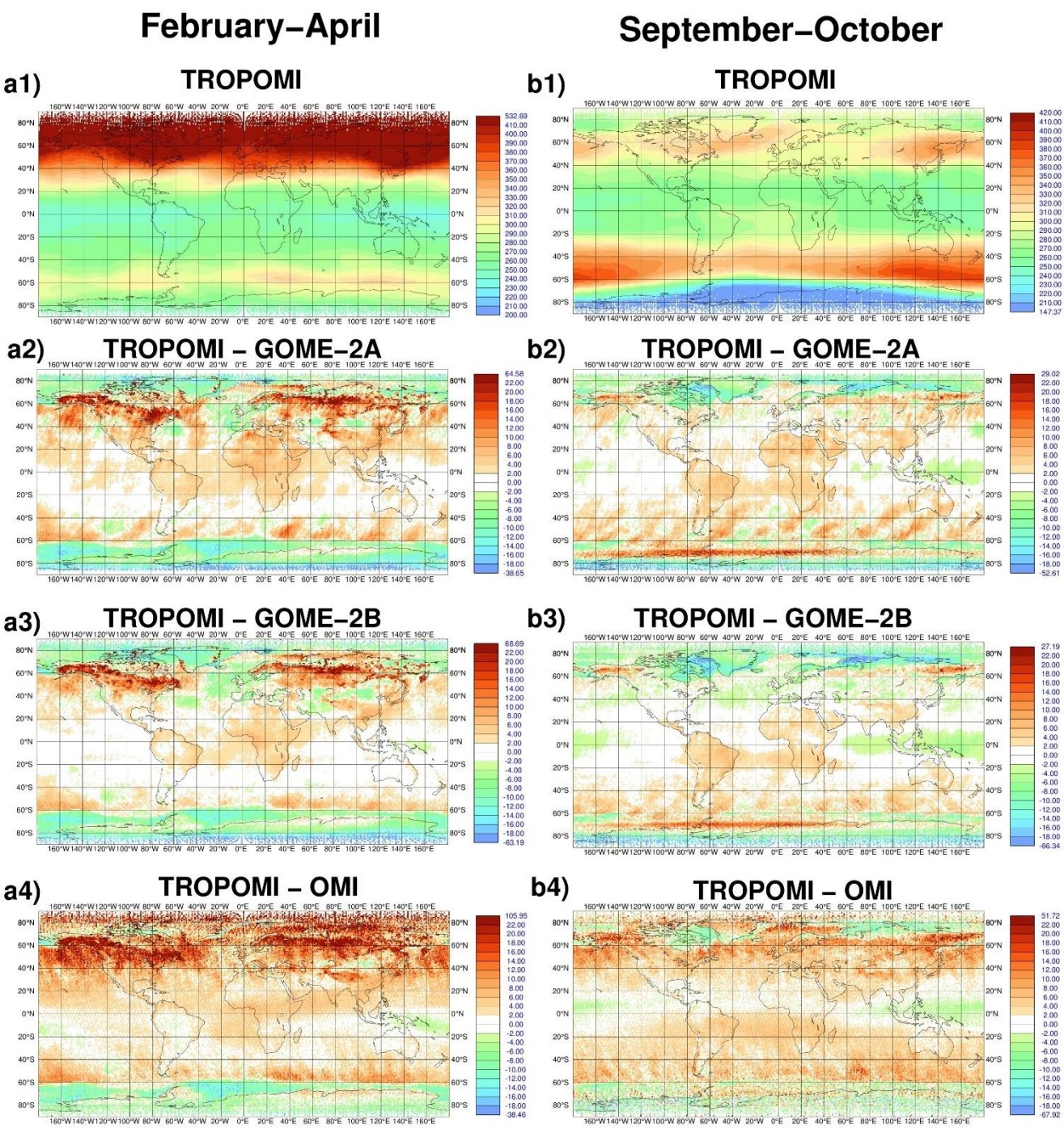

**Figure 3: Averages over (a) FMA 2018 and (b) SON 2018. Shown are (1) mean TROPOMI TCO3 fields, differences between (2) TROPOMI minus GOME-2A, (3) TROPOMI minus GOME-2B and (4) TROPOMI minus OMI. Shown are 'good' data. All values in DU.**

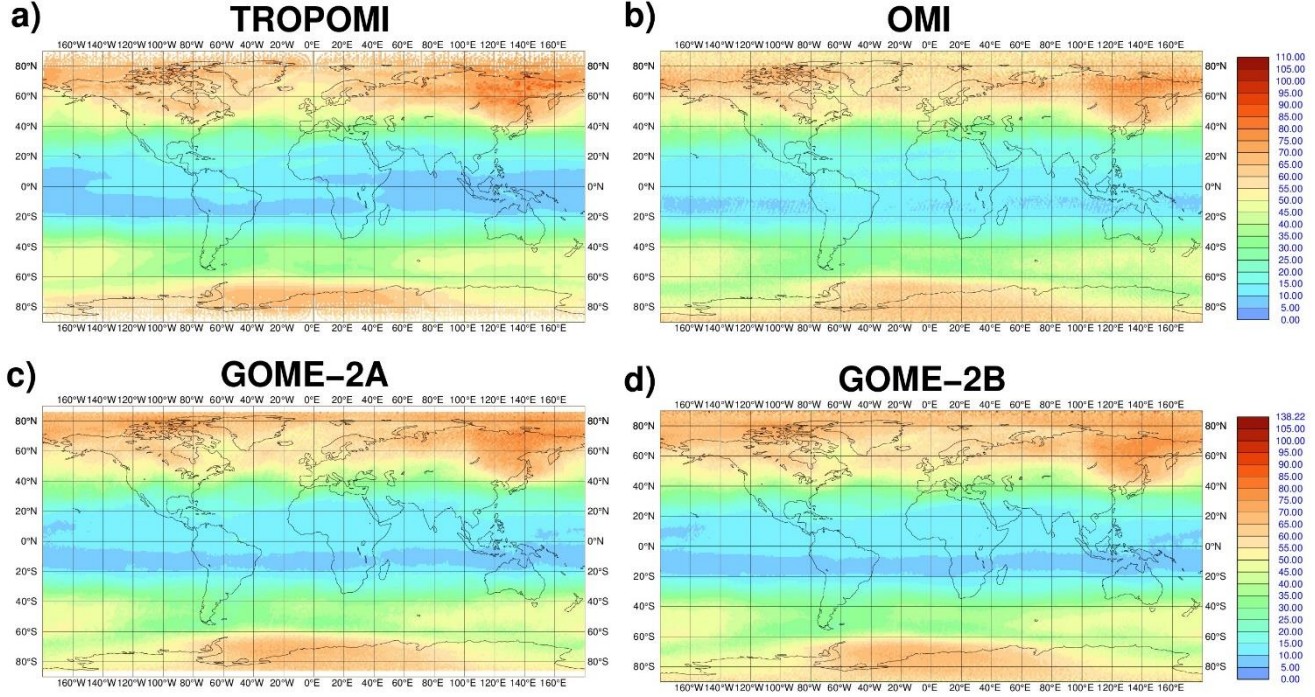

**Figure 4: TCO3 standard deviation of 'good' data for the period 26 November 2017 to 30 November 2018 from (a) TROPOMI, (b) OMI, (c) GOME-2A and (d) GOME-2B in DU.**

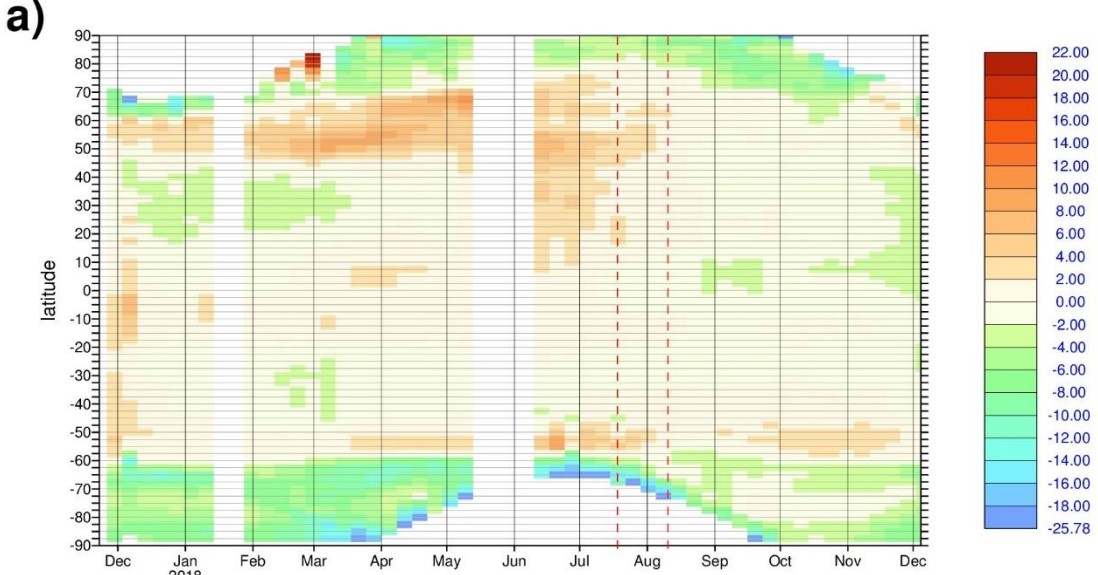

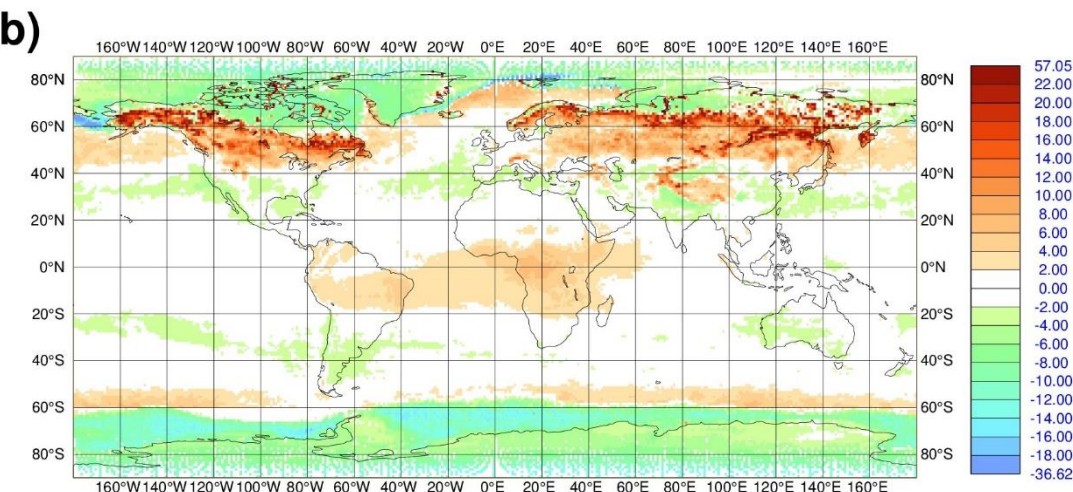

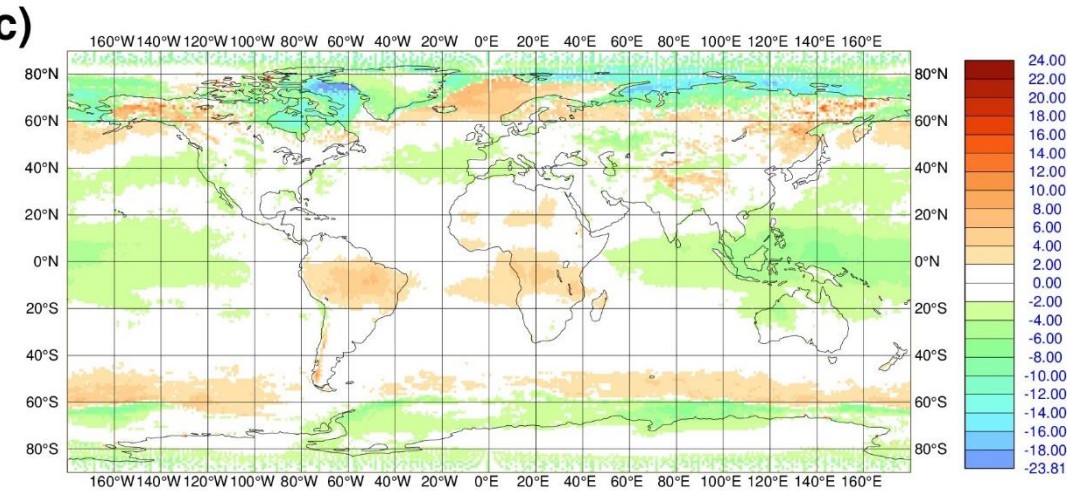

**Figure 5: (a) Timeseries of zonal mean weekly averaged TROPOMI TCO3 analysis departures (observations minus analysis) for the period 26 November 2017 to 30 November 2018. The red dashed lines mark changes in the retrieval versions (see Table 2). (b) Mean TROPOMI TCO3 analysis departures averaged over FMA 2018 and (c) mean analysis departures averaged over SON 2018 (compare with Figure 5). 2018. Shown are 'good' data. All values in DU. Values smaller than 2DU are white.**

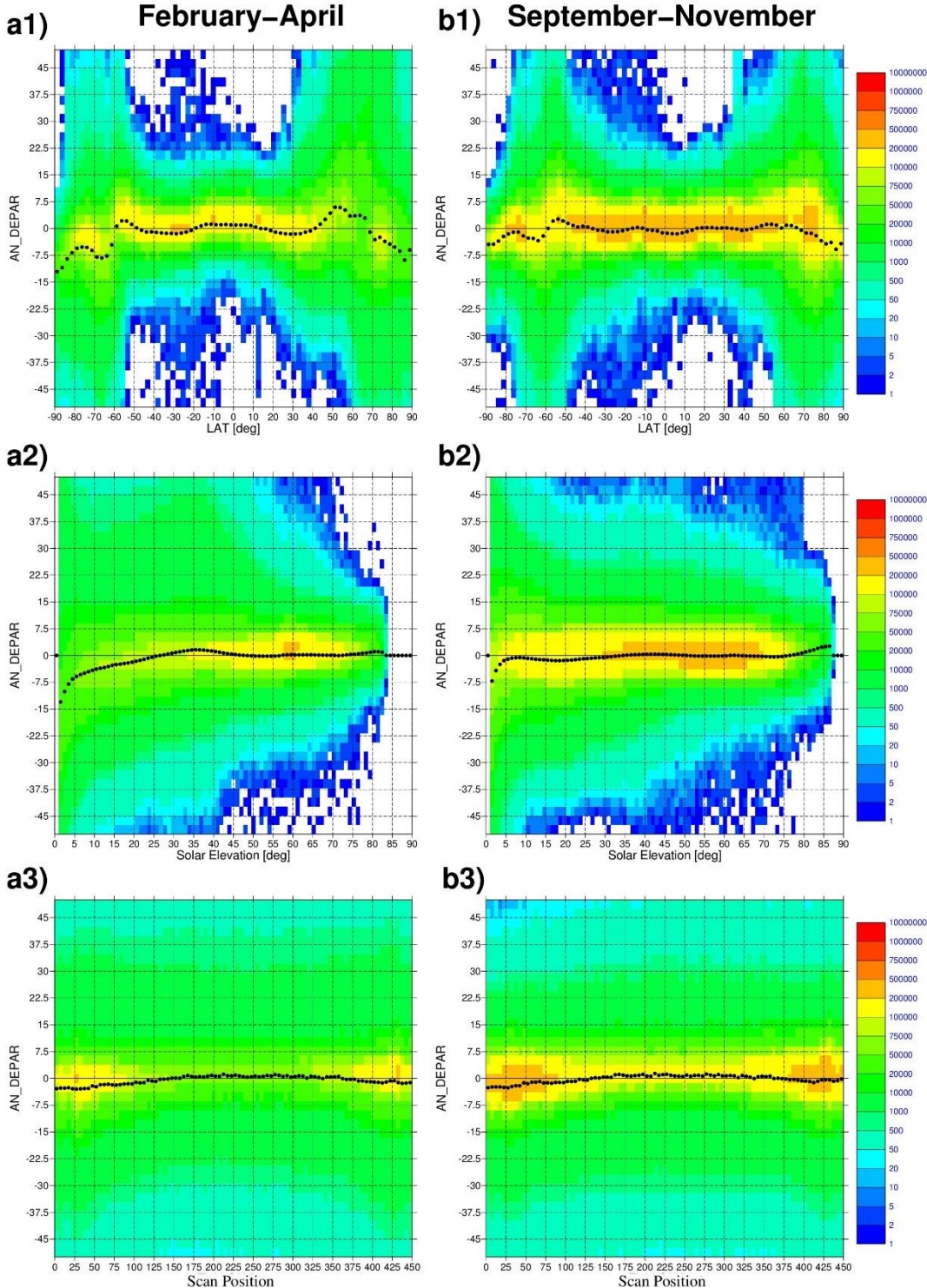

**Figure 6: Scatter plots of 'good' TROPOMI NRT TCO3 analysis departures against (1) latitude, (2) solar elevation and (3) scan position for (a) FMA 2018 and (b) SON 2018. Values are in DU.**

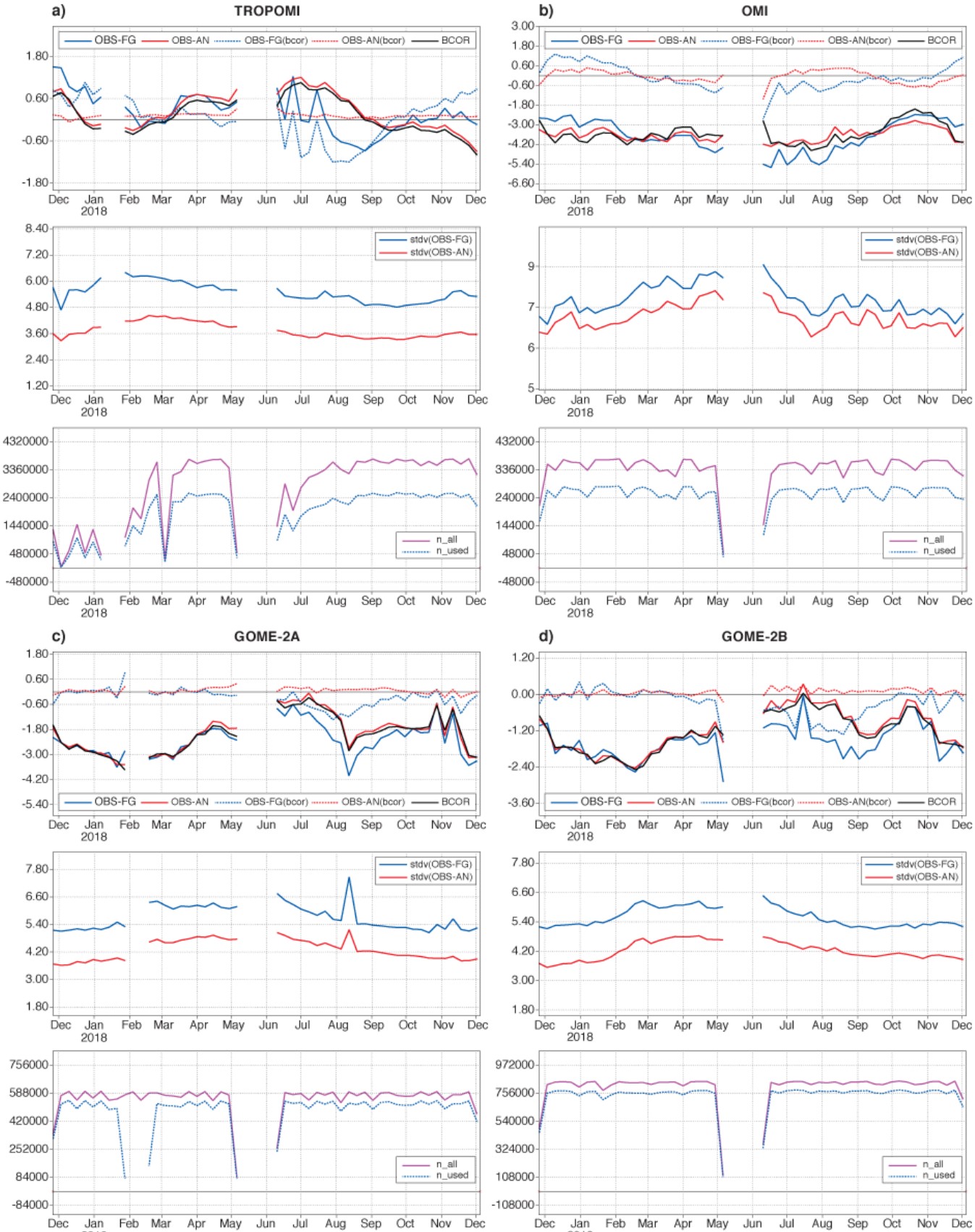

**Figure 7: Row 1: Time series for the period 26 November 2017 to 30 November 2018 of global mean weekly averaged TCO3 first-guess (solid blue) and analysis departures (solid red), bias corrected first-guess (dotted blue) and analysis departures (dotted red) and bias correction (black) in DU; row 2: standard deviation of first-guess (blue) and analysis departures (red) in DU and row 3: number of data (used in blue, all in magenta) from (a) TROPOMI, (b) OMI, (c) GOME-2A and (d) GOME-2B. Shown are 'used' data.**

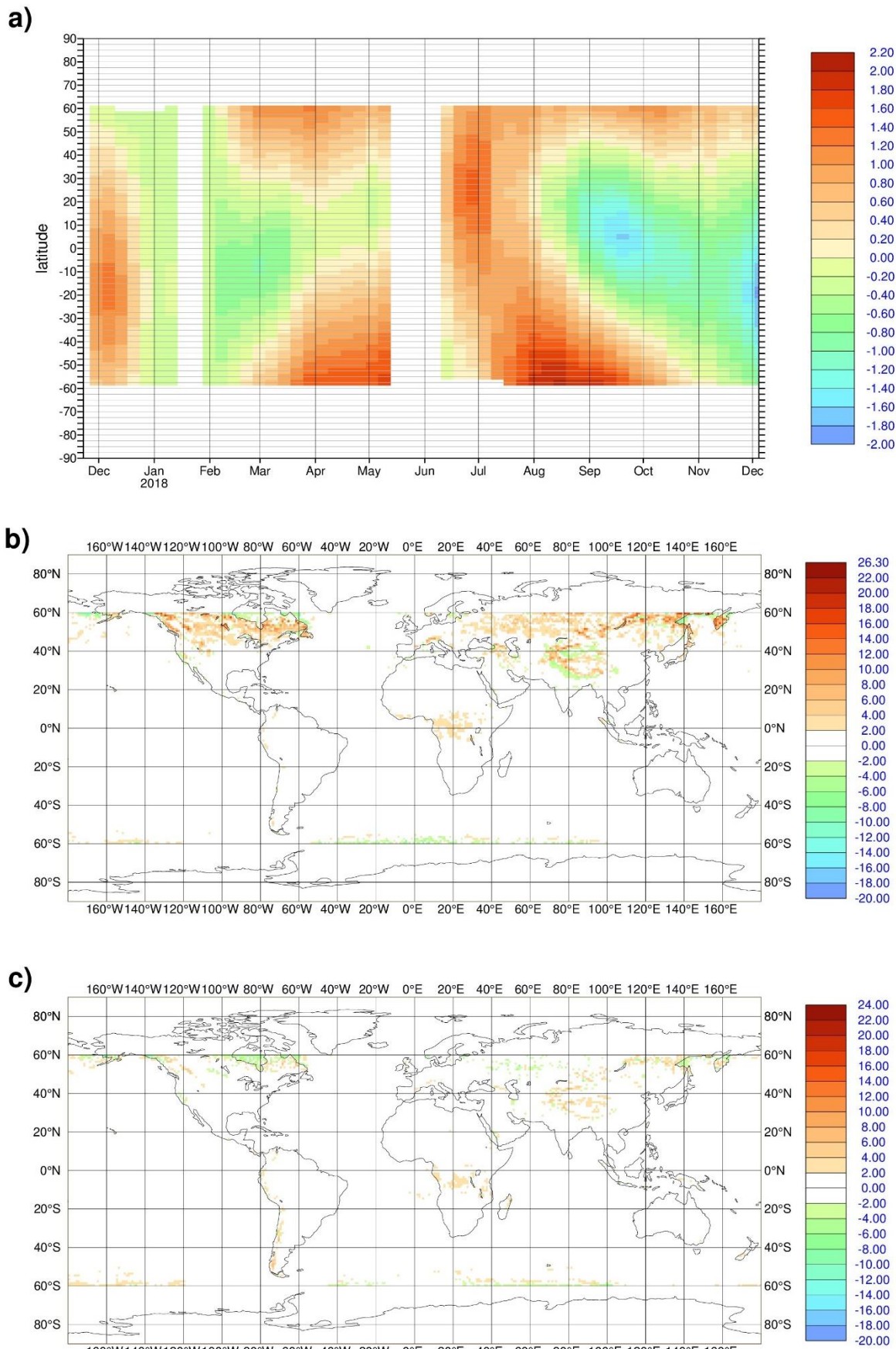

**Figure 8: (a) Timeseries of zonal mean weekly average bias correction applied to TROPOMI TCO3 for the period 26 November 2017 to 30 November 2018 and mean bias corrected analysis departures for (b) FMA 2018 and (c) SON 2018. All values in DU.**

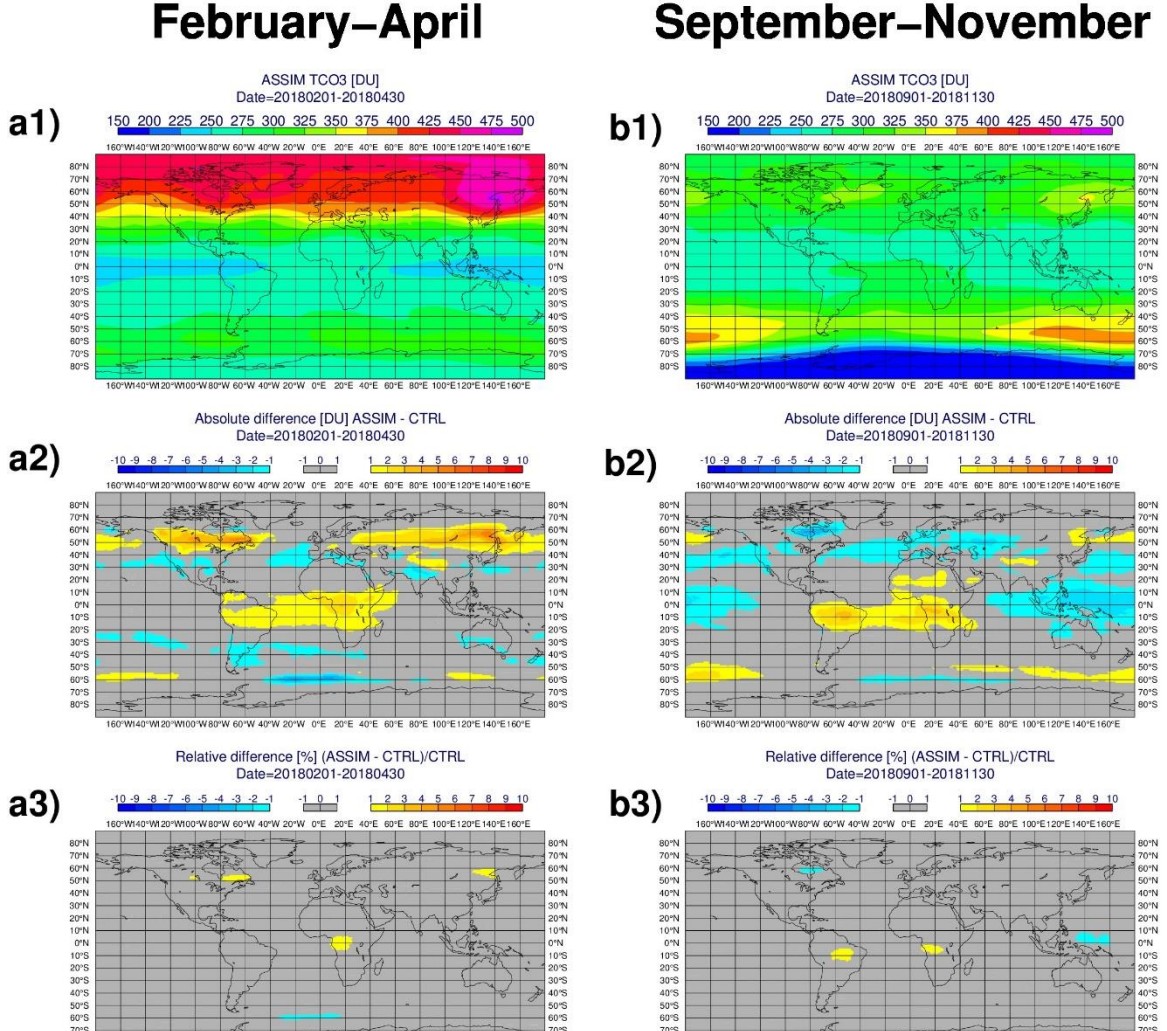

**Figure 9: Average fields for (a) FMA and (b) SON 2018. Shown are (1) mean TCO3 analysis from ASSIM, (2) the absolute differences between ASSIM and CTRL in DU and (3) the relative differences between ASSIM and CTRL in %.**

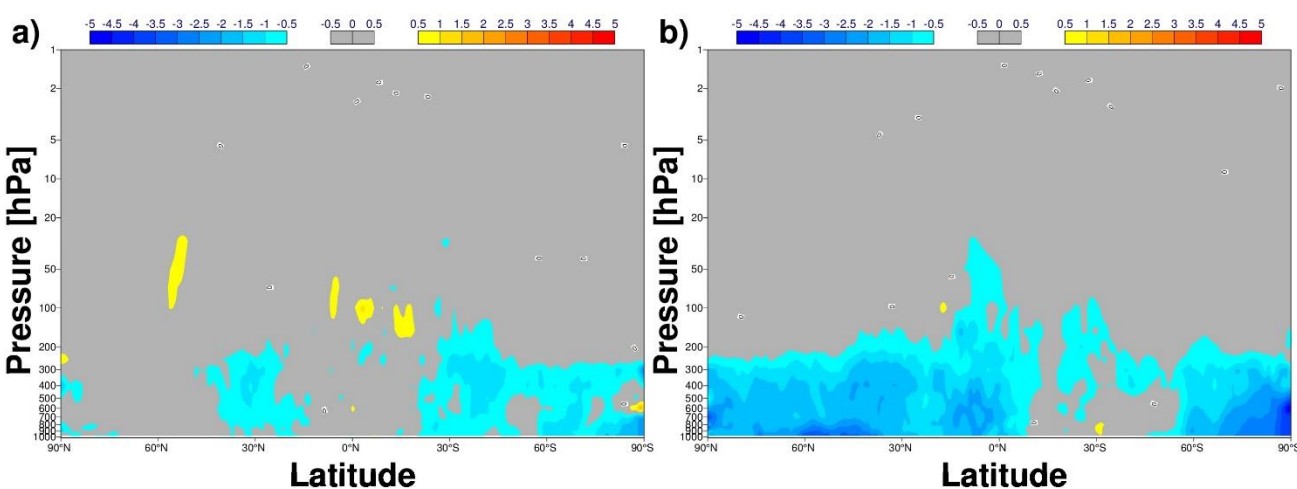

**Figure 10: Cross section of relative zonal mean O₃ mixing ratio differences from ASSIM minus CTRL averaged over (a) FMA 2018 and (b) SON 2018 in %.**

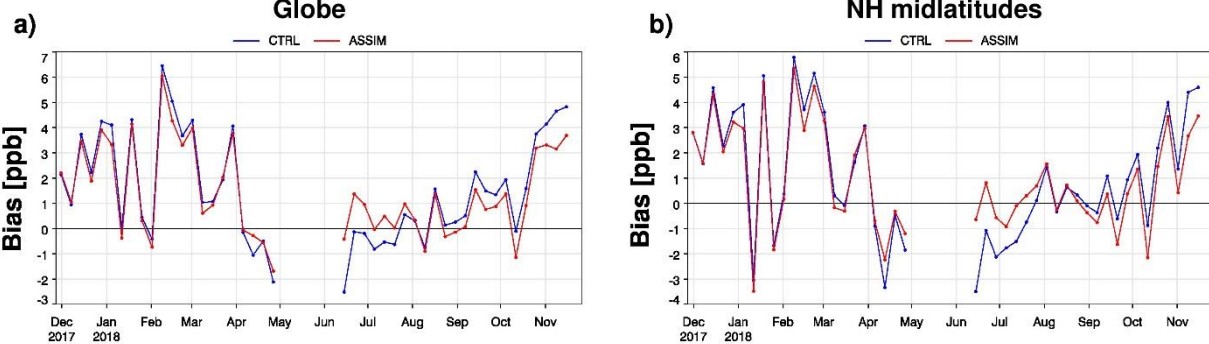

**Figure 11: Timeseries of weekly averaged TCO3 bias in DU from ASSIM (red) and CTRL (blue) compared to WOUDC Brewer data averaged over (a) the Globe (between 33 and 15 sites) and (b) NH midlatitudes (between 19 and 12 sites).**

a) **Globe**
b) **Arctic**
c) **NH midlatitudes**
d) **Tropics**
e) **SH midlatitudes**
f) **Antarctic**

**Figure 12: Mean relative O₃ bias for FMA 2018 in % between ASSIM (red) and CTRL (blue) and ozone sondes averaged over the (a) Globe, (b) Arctic, (c) NH midlatitudes, (d) Tropics, (e) SH midlatitudes and (f) Antarctic.**

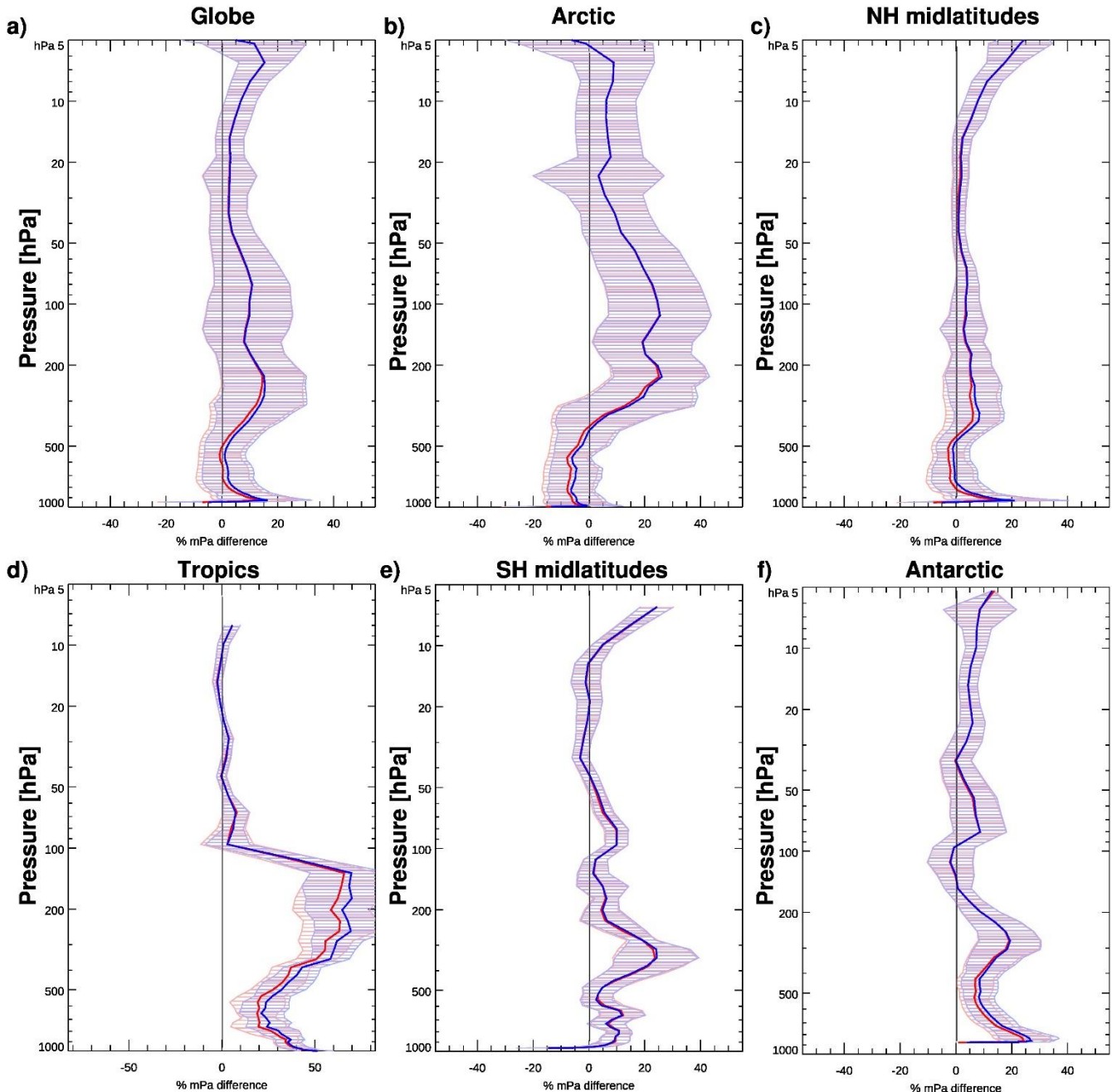

**Figure 13: Mean relative O₃ bias for SON 2018 in % between ASSIM (red) and CTRL (blue) and ozone sondes averaged over the (a) Globe, (b) Arctic, (c) NH midlatitudes, (d) Tropics, (e) SH midlatitudes and (f) Antarctic.**

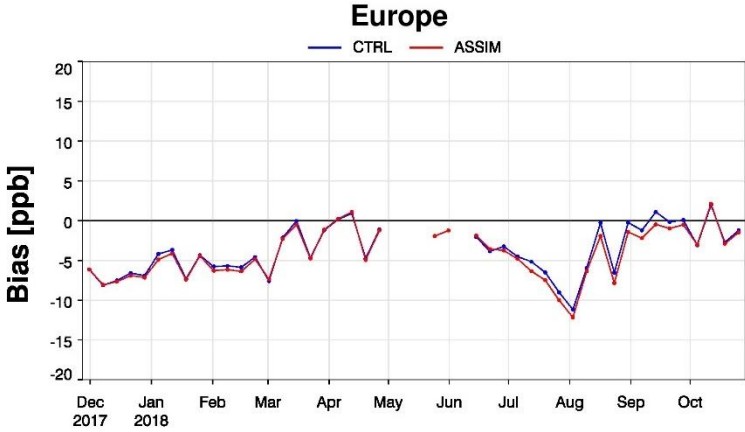

**Figure 14: Mean relative O₃ difference in % of ASSIM minus IAGOS aircraft data (red) and CTRL minus IAGOS (blue) for (a) FMA and (b) JAS 2018 averaged over (1) European airports, (2) SE Asian airports and (3) West African airports.**

**Figure 15: Timeseries of weekly averaged O3 bias in ppb from ASSIM (red) and CTRL (blue) compared to GAW surface data averaged over Europe (between 5 and 3 sites).**

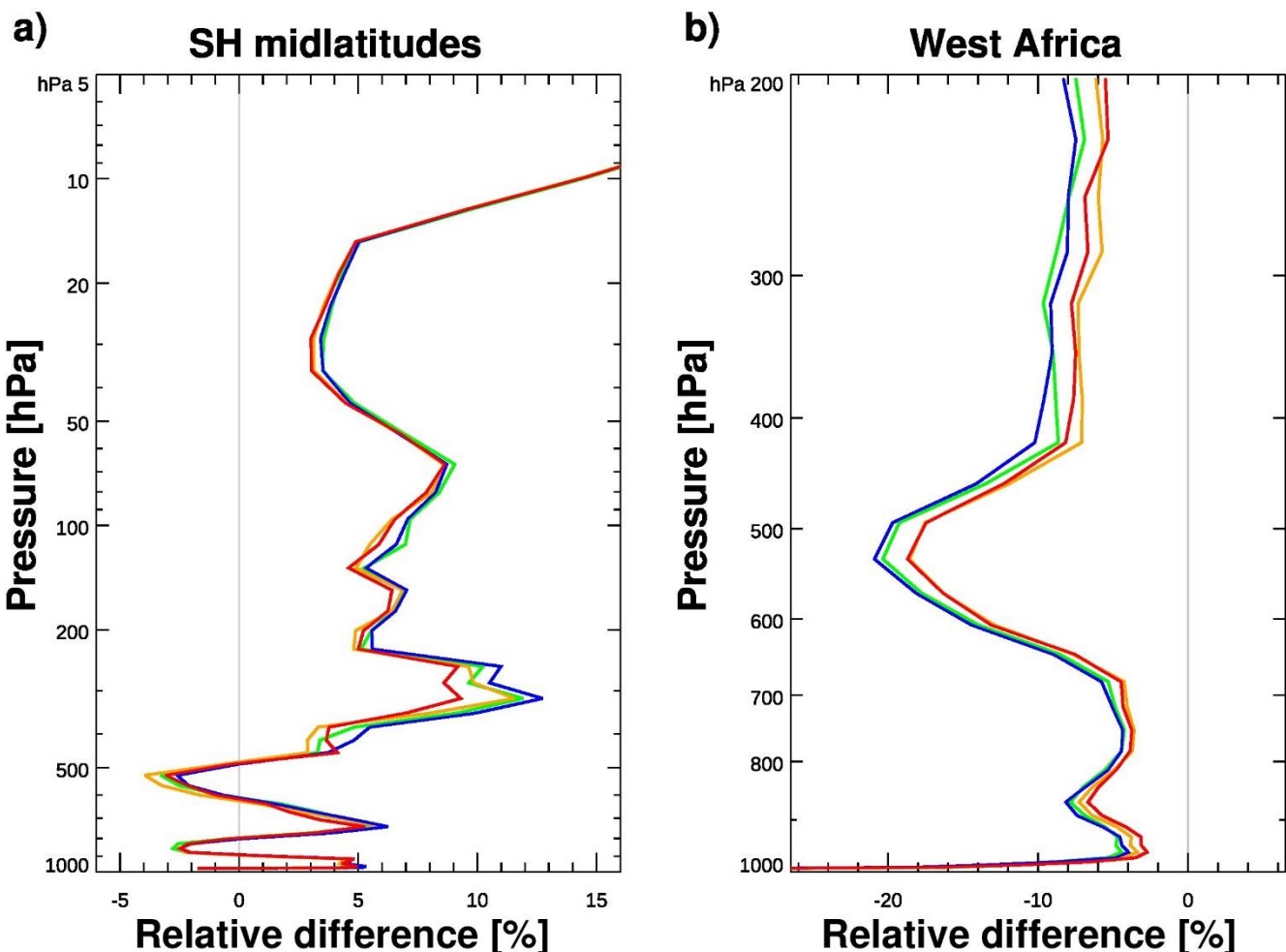

**Figure 16: Mean relative O$_3$ difference in % for FMA 2018 against (a) ozone sondes in SH midlatitudes and (b) IAGOS aircraft data at West African airports from ASSIM (red), ASSIM-OMI (orange), CTRL (blue) and CTRL-OMI (green).**

