# Peer review of "Monitoring and assimilation tests with TROPOMI data in the CAMS system: Near-real time total column ozone"

_Atmospheric Chemistry and Physics, 2018_

## Referee Comment (RC1) · Anonymous Referee #1 · 20 Nov 2018

Review of

"**Monitoring and assimilation tests with TROPOMI data in the CAMS system. Part 1: Near-real time total column ozone**"

By Inness et al., ACPD 2018

This manuscript describes the first testing of TROPOMI TCO3 NRT retrievals version v1.0.0 as provided by DLR in the CAMS system. They describe in detail the actual testing procedure that has been performed so far with intention to arrive at operational application of the TROPOMI O3 data within the CAMS system.

Even though significant biases have been detected in this early retrieval product, the authors show that once applied in the CAMS data-assimilation system the automatic bias correction scheme is able to remove a large portion of these biases, which then brings the TROPOMI data within magnitudes such that no deterioration, and even a small improvement is obtained when compared to independent observations. Also this configuration shows the strength of having the monitoring capabilities in the CAMS system to be able to quickly assess potential issues in the retrieval product (already during commissioning phase). This is valuable information to the retrieval product developers.

I agree such careful assessment of any retrieval product is a very valuable application of the CAMS system. However, the scientific interest of current testing may be limited, considering that the authors have clearly used a preliminary retrieval product (details, as will described in Loyola et al., 2019, are still "in preparation"), which is not expected to be used within the community. Hence, their assessment, even though very relevant for the developers of the retrieval algorithm and for its potential impacts when applied operationally within CAMS, may be of limited general interest to deserve documentation in a scientific publication of this kind. As result, the number of figures could possibly be reduced in the main text, and shifted to supplementary material, with exception of a selection of key figures, for which the figure quality could be improved. Furthermore, the evaluation period (effectively only about three months of data) is really short and also, for instance, not including the austral spring, which makes it hard to make an overall judgement.

Consequently. the description of the procedure to introduce new satellite instruments in the CAMS system is still valuable information, but may fit better in a GMD-type of publication.

Furthermore, the authors conclude that the TROPOMI data do not degrade the ozone in the current configuration and therefore are intending to include full assimilation of TROPOMI NRT TCO3 data in the operational CAMS analysis soon. Their argument is that this adds redundancy and resilience in case that some of the older instruments stop working. Here I have severe problems. It is true that the significant biases, as visible from the current, preliminary, version of the retrieval algorithm are more or less efficiently removed in the full data-assimilation system, but this is exactly owing to the presence of the existing observation system, including, for example, MLS and OMI onboard of AURA, which has already exceeded its foreseen lifetime. When these observations stop to be delivered, the bias correction algorithm may not work as efficiently, giving more weight to the (currently erroneous) TROPOMI retrievals. The magnitude of this effect has not been evaluated. Hence, to validate their statement that

adding TROPOMI with its current retrieval algorithm indeed adds the necessary resilience I suggest the authors to include two additional sensitivity experiments:

- One where observations from (for instance) the AURA instrument (MLS and OMI) are no longer included, and also no TROPOMI data is included
- One without the AURA observations, but now including TROPOMI observations to take over.

Such a study would indeed make a clearer case as to resilience of the system in the case of a sudden failure of these important instruments in the CAMS system, which would be a valuable assessment of the CAMS system that is also of quite more scientific interest.

**More detailed comments**

p1, l. 19 "agree well": I do not agree here: the TROPOMI data show biases wrt the CAMS system of up to 50 DU locally. Please reformulate.

p1, l.28 "less" should be "more"?

Abstract: this is too lengthy to my taste, and authors should try to condense. It doesn't need to be complete.

P1, l.39 (and also p11, l6): "After more tests": which kind of testing is meant here?

P2, l.34: the authors describe here the importance of the ozone hole in the earth system. This subject is however not covered in the assessment of the ozone assimilation experiment. The issue is that the reader is directed to wrong expectations.

P3, l.7: the authors discuss the important issue of resilience of the system. However, in my opinion they don't fully test this in the current manuscript, see my comments above. Please consider revision of statements, or inclusion of additional sensitivity experiments.

P3, l.22: "Because the departures …": I don't understand this sentence, please consider re-formulation.

Introduction: While authors make the case for the importance of ozone assimilation in CAMS, I miss references to other examples where new satellite data has been tested in early phase, e.g. in terms of composition, or possibly meteorology.

P3, l.31: "are not too large": this sounds like a very subjective statement. Can this be quantified in any way?

P4, l.3: for more clarity, suggest to rewrite "… as originally implemented in…" or "… as originating from the Chemistry…"

P4, l.30: here the authors state why the application of averaging kernels is not needed, either because they are not needed (having profile data as in MLS) or not available. This I find questionable, and I wonder if averaging kernel aspects could be important to reconciliate differences found between, for

instance, the GOME-2 and OMI retrievals as seen from Figure 5. At least, it would be good if the authors can back-up their statements with literature on the subject.

P6, l.12 "that all four instruments agree well": I find this statement with the current dataset a bit problematic, particularly when authors note differences in total columns of "up to 60 DU" (p6, l17). Please consider re-formulation

P6, l.30: "Tropics","problem with the OMI retrievals": The signal appears actually mostly isolated over the tropical continents. Therefore, couldn't it be a sign of something physical, associated to tropospheric chemistry, which is indeed picked up in the OMI retrieval and not in GOME-2, e.g. because of different sensitivities towards the lower altitudes? Evaluation with independent data shows indeed a negative model bias over West Africa (Fig. 16), although the model also shows positive bias averaged over the tropics when evaluated against sondes (Fig. 15). As bottom line, are you confident that the bias correction functioning properly when bias is removed from OMI and not from GOME?

P7, l.26: "they are less than 1% in the NH and SH", but this includes compensating errors, or? Please comment.

P7, l.37: about "60S and 50N" should be "about 60S and 40N".

P8, l.2: "destriping correction": although indeed a sensitivity to the scan position is shown, I find it confusing to refer to the 'destriping correction' here. I so far understood that this de-striping has been introduced to handle, for instance, row-anomaly issues with particular affected rows, as is the case with OMI, but not so much with changing signal towards the edges of the scan, which may require a more physical explanation in the retrieval algorithm. Can you comment?

P8, l.10, "blacklisting poleward of 60 degree". Seeing the emerging biases at 40 degree north, it would make sense to blacklist data from this latitude onwards, or?

P9, l.1, "are less than 2%". This refers to the average bias for the 26 Nov – 3 May time period. But as is clear from Figure 10, there is only little data in the December-January time period, which means that the difference for this period with/without TROPOMI data is negligible. Therefore, I believe it is more meaningful to present & report these biases for the 28 January – 3 May period, as done also in Figure 13.

P9,l.5: "26 November": Figure caption of Fig. 13 writes " 28 January". Please check.

P9, l.10: "the impact would be larger": Yes, I agree, and this is worrying, particularly as the positive impact, seen so far, may no longer be the case. Could you please comment?

P9, l.35 "observation**s**"

P9, l.39: "Loyola et al., 2019": the current data-stream appears preliminary, considering that the algorithm description has not been published yet. This is important to stress.

P10, l.20 "solar elevation": this seems only partly an appropriate predictor. Would the CAMS system be flexible enough to add different predictors in the system for its bias correction scheme, such as albedo?

P10, l.34 "to add redundancy", This is actually not tested, please see my comments above. Please comment.

Figure 10: Figure quality appears insufficient for publication, and legend seems incomplete. Please consider improved legibility.

---

## Referee Comment (RC2) · Anonymous Referee #2 · 4 Dec 2018

The paper describes the outcomes of the assimilation tests in the CAMS system with Tropomi TCO3 v1.0.0 data. The paper properly explains the methods as well the input data used in this work. Presentation is clear and well structured. Proper acknowledgements are given to all authors as well as other data providers/sources.

The paper describes how CAMS assimilation system may be used to monitor bias as well as other measurement characteristics of a new instrument and thus, show how CAMS system may be used as an independent source of quality control. This may benefit both algorithm developers as well as data users.

The basic idea behind this work is not novel because the CAMS assimilation system

has existed quite long time already. However, the Tropomi data itself is very interesting and can be seen as a step forward for the new instruments in future, like onboard EPS-SG platforms. The Tropomi instrument is in the afternoon orbit and thus, fulfills nicely data retrieved with morning instruments, like GOME-2 and other instruments in morning orbits. Thus, the topic of this paper is interesting indeed.

The paper show results over period 26th November to 3rd May 2018. Most of the figures show averaged values over the whole data period. However, the figures (like figure 7) indicate problems in L1 and L2 in December-January, which may be due to the early commissioning phase. Thus, the first data samples may not be at the same quality level than in Feb-April, which may have an effect to information content of the images showing averages of the whole period. Have you checked this? Thus, it would be interesting and useful to see a little bit more information about assimilation and control fields at certain fixed time steps.

Furthermore, there are some other concerns at the same time. The first one is the length of the assimilation period from about December to April with several larger data caps. The data of about three months may not be well representative over the seasonal cycles, for example. Therefore, it is difficult to make conclusions beside February-April for the rest of the year.

The second concern is about the TCO3 version v1.0.0. The current version is already 1.01.02 with improved OCRA and ROCINN as described by Loyola et al 2018. The retrieval algorithms for Tropomi are under fast development right now and thus, the results shown in this paper may be somewhat outdated already by now.

Page 9, line 9-10 and page 10, lines 34-35: It's mentioned that if no other data is available, the effect of Tropomi data would be larger. This seems to be a justified conclusion indeed. Furthermore, according the paper, the main reason to assimilate Tropomi TOC3 v 1.0.0 and not to wait more mature data version is that the assimilation of Tropomi data as soon as possible would be beneficial in case of failure of older instruments. However, to evaluate this properly, there should be assimilation tests where some other instruments are removed from the system. Otherwise, it may be difficult to support the argument because it's difficult to predict how CAMS model behaves when some other instruments are removed. I'm not expert in CAMS data assimilation and thus, some more evidence could be presented.

Several paragraphs/sentences associated to certain figures (like 4, 8, 11 and 13 etc) are very short. Thus, the analysis (text) and the figures don't seem to be in balance. Therefore, I would suggest that the authors should reconsider the structure of the paper and put some of the paragraphs together, perhaps. Also, it could be useful to reconsider if all of those figures are necessary.

Detailed comments:

Table 1: What is OMPS data version?

Page 3, line 20: It is difficult to see this as long term monitoring because the actual data period is about three months.

Page 4, line 8-10: A very short second paragraph in 2.1. Is it possible to merge this with the first paragraph in 2.1?

Page 6, line 18: "..is mainly lower..". This seems to be too general conclusion because the high max areas reach the latitude 65. There seems to be clear land-sea separation.

Page 6, line 41: It would be better to use the version with the new climatology in this study.

Page 7, line 1: "…small negative departures elsewhere…". There seems to be quite large negative departure over Antarctica but it's not mentioned here or before?

Figure 7, tables and texts: The latitudes are not consistent through the text. In fig 7 they are -70 – +70 whereas in tables 2 and 3 as -90 - +90. Furthermore, the active assimilation was done only within the latitude band of -60 - +60 and thus, the results in

the Fig 7 could be different if the latitudes were restricted accordingly. Thus, the reason for different latitude bands should be explained (some are clear, some are not so).

Page 7, line 30: "Small" is subjective term here. For example, 55N, the departure is about 10 DU. In general, to use 60N as a separation seems to be a little bit problematic because in several figures the actual separation could be 50N or 55N.

Figure 11 and Page 8, line 31-35: Is the swath angle dependency seen in tropics in the fig 11b?

Fig 12 b: Perhaps scale from -5 to 5 DU with 0.5 DU tics could work better.

Page 9, line 33: Interesting to see this clear improvement

---

## Author Response (AR1)

**Replies to reviewers**

We have changed the paper considerably by including additional TROPOMI data so that we now cover the period 26 November 2017 to 30 November 2018. This addresses some of the criticism from both reviewers and will hopefully make our study a better paper. We now show timeseries for the whole period as well as averages over February-May and September-November 2018.

We have addressed all comments from the two reviewers and have added our replies in red below.

**Reviewer 1**

5

This manuscript describes the first testing of TROPOMI TCO3 NRT retrievals version v1.0.0 as provided by DLR in the CAMS system. They describe in detail the actual testing procedure that has been performed so far with intention to arrive at operational application of the TROPOMI O3 data within the CAMS system.

Even though significant biases have been detected in this early retrieval product, the authors show that once applied in the CAMS data-assimilation system the automatic bias correction scheme is able to remove a large portion of these biases, which then brings the TROPOMI data within magnitudes such that no deterioration, and

15 even a small improvement is obtained when compared to independent observations. Also this configuration shows the strength of having the monitoring capabilities in the CAMS system to be able to quickly assess potential issues in the retrieval product (already during commissioning phase). This is valuable information to the retrieval product developers.

I agree such careful assessment of any retrieval product is a very valuable application of the CAMS system.

- 20 However, the scientific interest of current testing may be limited, considering that the authors have clearly used a preliminary retrieval product (details, as will described in Loyola et al., 2019, are still "in preparation"), which is not expected to be used within the community. Hence, their assessment, even though very relevant for the developers of the retrieval algorithm and for its potential impacts when applied operationally within CAMS, may be of limited general interest to deserve documentation in a scientific publication of this kind. As result, the
- 25 number of figures could possibly be reduced in the main text, and shifted to supplementary material, with exception of a selection of key figures, for which the figure quality could be improved. Furthermore, the evaluation period (effectively only about three months of data) is really short and also, for instance, not including the austral spring, which makes it hard to make an overall judgement.

The early data are publicly available and are expected to be used by the community. They are not preliminary data and were processed with the same NRT algorithm as the normal NRT data after June 2011, only they were 're-processed' with this algorithm, i.e. not produced in NRT. We state that they were 're-processed with NRT algorithm' to stress that they were not produced with the offline algorithm. We have re-phrased this in the document and hope we make this clearer now.

The retrieval was well documented in the ATBD (http://www.tropomi.eu/documents/atbd, last access Jan 2019), which was last updated in October 2018. However, due to other priorities a respective manuscript Loyola et al. (2019b) has not been published yet.

S5P/TROPOMI Total Ozone ATBD Rob Spurr, Diego Loyola, Michel Van Roozendael, Christophe Lerot, Klaus-Peter Heue, Jian Xu: S5P-L2-DLR-ATBD-400A, issue 1.6, 2018-10-17

We now state in section 2.2:

40 "For the work in this paper we use NRT TROPOMI TCO3 data (Loyola et al.; 2019b). These include TROPOMI data (V1.0.0) for the period 26 November 2017 to 3 May 2018 V1.0.0 that were reprocessed with the NRT algorithm and NRT TROPOMI data V1.0.0-V1.1.2 for the period 11 June to 30 November 2018 (see Table 2). No data were acquired at ECMWF from 4 May to 10 June 2018 for technical reasons. The TROPOMI TCO3 retrieval is based on the GDP 4.x algorithm original developed for GOME (van Roozendael et al., 2006), adapted to SCIAMACHY (Lerot et al., 2009) and further improved for GOME-2 (Loyola et al., 2011; Hao et al., 2014). The major TCO3 algorithm updates for TROPOMI compared to the heritage algorithms used for GOME-2 are the more precise treatment of clouds as scattering layers (Loyola et al., 2018), an optimized wavelength for the calculation of air mass factors

- 5 (328.2 nm instead of 325.5 nm), better a-priori ozone profile information (including the tropospheric climatology by Ziemke et al. (2011)) and a destriping correction. This destriping correction was introduced because total vertical ozone columns showed small striping structures. The correction factor is based on the ratio between the mean for individual rows and the mean for all rows over a certain region and period. We averaged the total columns within the tropics for January to April 2018 for both all 450 rows individually and over all 450 rows,
- 10 resulting in an array of 450 numbers, ranging between 0.99 and 1.015. Multiplying the VCD with the correction factor changes the result by about ±1%. The correction factor has been rechecked but no update seemed necessary up to now. The TROPOMI retrieval is described in the S5P/TROPOMI Total Ozone ATBD (Spurr et al., 2018)."

Consequently. the description of the procedure to introduce new satellite instruments in the CAMS system is still valuable information, but may fit better in a GMD-type of publication.

Furthermore, the authors conclude that the TROPOMI data do not degrade the ozone in the current configuration and therefore are intending to include full assimilation of TROPOMI NRT TCO3 data in the operational CAMS analysis soon. Their argument is that this adds redundancy and resilience in case that some of the older instruments stop working. Here I have severe problems. It is true that the significant biases, as

- 20 visible from the current, preliminary, version of the retrieval algorithm are more or less efficiently removed in the full data-assimilation system, but this is exactly owing to the presence of the existing observation system, including, for example, MLS and OMI onboard of AURA, which has already exceeded its foreseen lifetime. When these observations stop to be delivered, the bias correction algorithm may not work as efficiently, giving more weight to the (currently erroneous) TROPOMI retrievals. The magnitude of this effect has not been evaluated.
- 25 Hence, to validate their statement that adding TROPOMI with its current retrieval algorithm indeed adds the necessary resilience I suggest the authors to include two additional sensitivity experiments:

One where observations from (for instance) the AURA instrument (MLS and OMI) are no longer included, and also no TROPOMI data is included. One without the AURA observations, but now including TROPOMI observations to take over.

30 Such a study would indeed make a clearer case as to resilience of the system in the case of a sudden failure of these important instruments in the CAMS system, which would be a valuable assessment of the CAMS system that is also of quite more scientific interest.

TROPOMI will not be able to compensate for the loss of MLS O3 profiles which are very important in the CAMS ozone analysis (and we now explicitly state in this in the paper). We only claim TROPOMI will be a good

- 35 replacement for the currently used total column data (i.e. OMI or one of the GOME-2 instruments). We therefore do not think a test with/without MLS data is meaningful or within the scope of this paper. It addresses a completely different issue, i.e. the performance of the CAMS system when not assimilating height resolved O3 profile data. We know that without MLS the CAMS analysis can struggle to reproduce well the vertical ozone distribution.
- 40 We agree that we have not shown that TROPOMI can replace OMI (or GOME-2AB) one for one. The reason is that this paper is not supposed to be an observing system experiment paper, where we test the impact of each individual data set, but describes the inclusion of the new data into an existing system. As the CAMS system is an operational system it is always possible that data sets are not available for limited periods or instruments die and the analysis will change.
- 45 Because of time and limited computer resources it was not possible to run two further experiment for the whole period covered in this paper. We have however run the 2 suggested experiments for December 2017-April 2018 and now include an additional Figure (new Fig 16) that shows comparisions of these 2 new experiments, ASSIM

and CTRL with ozone sondes for FMA 2018. We do indeed not see a big impact if TROPOMI data are used without OMI (if anything, excluding OMI seems to slightly improve the fit to IAGOS at West African airports). On the whole, the differences between the experiments are small. As the quality of the TROPOMI data is better in the later part of 2018 than during these first months we do not expect that their assimilation would degrade the CAMS analysis if one of the other instruments failed.

We have therefore kept the statement in the paper but have added at the end of section 3.2.2:

"On the whole, the impact of the TROPOMI assimilation in the CAMS system is relatively small because the CAMS analysis is already well constrained by the other data sets that are assimilated routinely which are a combination of TCO3 data (OMI, GOME-2AB), O3 layers (SBUV/2, OMPS) and O3 profiles (MLS) (see Table 1). If

- 10 no other O3 data were available and only TROPOMI TCO3 data were assimilated the impact on the CAMS O3 analysis would be larger. To confirm that TROPOMI could serve as a good replacement if one of the older TCO3 instruments (OMI, GOME-2AB) failed two further experiments were run for the period 26 November 2017 to 30 April 2018: one mimicking the configuration of CTRL, but without OMI (CTRL-OMI) and the other mimicking the configuration of ASSIM without OMI (ASSIM-OMI). Compared to ozonesondes and IAGOS data the differences
- 15 between these experiments and ASSIM and CTRL are very small indeed. The largest differences between the four experiments are found in the SH midlatitudes when compared with ozone sondes (Figure 16a) and over West African airports compared with IAGOS (Figure 16b). Even here, the differences between ASSIM and ASSIM-OMI are small and both fit the independent observations better than CTRL and CTRL-OMI. There is even a sign that removing OMI leads to a small improvement in the fit to IAGOS over West Africa. In all other areas the
- 20 differences between the experiments with and without OMI were negligible when compared to sondes or IAGOS. These findings agree with results from longer observation system experiments that were carried out with the CAMS system for the years 2013 and 2014 in a different context (not shown) which showed only small changes to the CAMS O3 analysis if one of the TCO3 instruments was removed confirming that the CAMS analysis is well constrained and that there is some redundancy in the system. We are therefore confident that TROPOMI will be
- 25 able to counterbalance the loss of one of the older TCO3 instruments. Removing MLS O3 profiles has a much larger (negative) impact on the CAMS O3 analysis (e.g. Flemming et al., 2011) and TROPOMI would not be able to replace the MLS profiles as it does not provided data with similar vertical resolution."

**More detailed comments**

p1, l. 19 "agree well": I do not agree here: the TROPOMI data show biases wrt the CAMS system of up to 50 DU 30 locally. Please reformulate.

**We now say:**

5

"The TROPOMI TCO3 data agree to within 2% with the CAMS analysis over large parts of the Globe between 60°N and 60°S and also with TCO3 retrievals from the Ozone Monitoring Instrument (OMI) and the Global Ozone Monitoring Experiment-2 (GOME-2) that are routinely assimilated by CAMS. However, the TCO3 NRT data from

35 TROPOMI show some retrieval anomalies at high latitudes, at low solar elevations and over snow/ice (e.g. Antarctica and snow-covered land areas in the Northern Hemisphere) where the differences with the CAMS analysis and the other datasets are larger. These differences are particularly pronounced over land in the NH during winter and spring (when they can reach up to 40 DU).."

p1, I.28 "less" should be "more"?

40 Yes, this has been corrected.

Abstract: this is too lengthy to my taste, and authors should try to condense. It doesn't need to be

complete.

**The abstract has been shortened.**

P1, I.39 (and also p11, I6): "After more tests": which kind of testing is meant here?

We have removed this statement as the data are by now actively assimilated in the operational CAMS system. This is now mentioned in the conclusion.

P2, I.34: the authors describe here the importance of the ozone hole in the earth system. This subject is however not covered in the assessment of the ozone assimilation experiment. The issue is that the reader is directed to wrong expectations.

The paper does now include data for the ozone hole period, so we have kept that sentence.

P3, I.7: the authors discuss the important issue of resilience of the system. However, in my opinion they don't fully test this in the current manuscript, see my comments above. Please consider revision of statements, or inclusion of additional sensitivity experiments.

10 Please see our comments to the more general statement above. We also changed the sentence to

"... system against the loss of any of the older instruments whose TCO3 retrievals are currently assimilated by CAMS...' to make clear that it would not be able to help with the loss of MLS.

P3, I.22: "Because the departures ...": I don't understand this sentence, please consider re-formulation.

We have rephrased this to: "Because the departures are smaller than the absolute observation values they show 15 up day-to-day changes better than when looking at the absolute model fields or observation values."

Introduction: While authors make the case for the importance of ozone assimilation in CAMS, I miss references to other examples where new satellite data has been tested in early phase, e.g. in terms of composition, or possibly meteorology.

The problem is that these early tests are usually not written up, because they are considered routine work. The

20 best ever at ECMWF was NOAA-16 AMSUA radiances that were assimilated just 5 weeks after launch but that was not written up as it was a routine operation. Probably the best example of completely new data in the ECMWF system would be AIRS where the system was trained with simulated data before launch to speed up the implementation. This is written up in https://onlinelibrary.wiley.com/doi/abs/10.1256/qj.04.171 but perhaps not entirely relevant for our study. As our study now uses a year worth of S5P data we do not think it is 25

necessary to include this reference.

P3, I.31: "are not too large": this sounds like a very subjective statement. Can this be quantified in any way?

**We have rephrased this to small**

P4, I.3: for more clarity, suggest to rewrite "... as originally implemented in..." or "... as originating from the Chemistry..." 30

**We have changed it to: "... as originally implemented in..."**

P4, I.30: here the authors state why the application of averaging kernels is not needed, either because they are not needed (having profile data as in MLS) or not available. This I find questionable, and I wonder if averaging kernel aspects could be important to reconciliate differences found between, for instance, the GOME-2 and

35 OMI retrievals as seen from Figure 5. At least, it would be good if the authors can back-up their statements with literature on the subject.

We have removed the sentence and now only state that averaging kernels are currently not used:

"Averaging kernels are currently not used for the assimilation of ozone retrievals in the CAMS system. It is planned to test their use in the future."

P6, I.12 "that all four instruments agree well": I find this statement with the current dataset a bit problematic, particularly when authors note differences in total columns of "up to 60 DU" (p6, I17). Please consider reformulation

We have rephrased the description of that Figure (now Fig. 3).

- 5 P6, I.30: "Tropics", "problem with the OMI retrievals": The signal appears actually mostly isolated over the tropical continents. Therefore, couldn't it be a sign of something physical, associated to tropospheric chemistry, which is indeed picked up in the OMI retrieval and not in GOME-2, e.g. because of different sensitivities towards the lower altitudes? Evaluation with independent data shows indeed a negative model bias over West Africa (Fig. 16), although the model also shows positive bias averaged over the tropics when evaluated against sondes
- 10 (Fig. 15). As bottom line, are you confident that the bias correction functioning properly when bias is removed from OMI and not from GOME?

The statement referring to a potential problem with the OMI retrievals has been removed.

P7, I.26: "they are less than 1% in the NH and SH", but this includes compensating errors, or? Please comment.

We have removed Figure 7 and instead now show a zonal mean timeseries of the differences in Figure 5 that
shows differences better by latitude. We also show mean biases and their standard deviations for 5 latitude
bands now (90-60N, 60-30N, 30N-30S, 30-60S, 60-90S) in Table 3 and 4 (were Table 2 and 3).

P7, I.37: about "60S and 50N" should be "about 60S and 40N".

We have re-done the Figures for FMA 2018 and SON 2018 and now say 60S-45N.

- P8, I.2: "destriping correction": although indeed a sensitivity to the scan position is shown, I find it confusing to refer to the 'destriping correction' here. I so far understood that this de-striping has been introduced to handle, for instance, row-anomaly issues with particular affected rows, as is the case with OMI, but not so much with changing signal towards the edges of the scan, which may require a more physical explanation in the retrieval algorithm. Can you comment?
- We do not observe a row anomaly up to now in the TROPOMI data, but besides that the reviewer is right the destriping correction was implemented to handle small scale variations that are observed for certain rows and are constant in time in principle it might also affect the edges. The row dependency found in the comparison to CAMS however is probably not caused by a systematic row dependency in the O3 columns. The TROPOMI data themselves are routinely quality controlled (http://mpc-l2.tropomi.eu/#o3, January 2019) and a systematic dependency was not observed.
- 30 The exact reason for the row dependency in the differences should be investigated in more detail; it might be caused by a shift in the position with the altitude. The maximum viewing zenith angle is 66.9° taking the earth's curvature into account the minimum elevation angle of the light being reflected to S5P is about 11.4°. At 12 km altitude, the difference to the central coordinate of the S5P ground pixel is about 59 km (1.5 CAMS grid cells). However, this does not explain why the ozone column decreases towards the edges, because the local column 35 might be both higher and lower. More work is needed to study this.

We have removed several references to the destriping correction from the manuscript and have added in section 2.2:

"This destriping correction was introduced because total vertical ozone columns showed small striping structures. The correction factor is based on the ratio between the mean for individual rows and the mean for all

40 rows over a certain region and period. We averaged the total columns within the tropics for January to April 2018 for both all 450 rows individually and over all 450 rows, resulting in an array of 450 numbers, ranging between 0.99 and 1.015. Multiplying the VCD with the correction factor changes the final result by about ±1%. The correction factor has been rechecked but no update seemed necessary up to now." P8, l.10, "blacklisting poleward of 60 degree". Seeing the emerging biases at 40 degree north, it would make sense to blacklist data from this latitude onwards, or?

Not necessarily. Because we see a lot of the problems at low solar elevations a blacklist based on latitude and SOE should work well. Also, the bias correction varies by latitude (because of the SOE parameter). Furthermore, the absolute TCO2 values are also larger path of 40N on the relative areas are still in a space where we see the

the absolute TCO3 values are also larger north of 40N so the relative errors are still in a range where we can try to use the data. The really bad outliers will be rejected by the first-guess check or given less weight by the variational quality control.

P9, l.1, "are less than 2%". This refers to the average bias for the 26 Nov – 3 May time period. But as is clear from Figure 10, there is only little data in the December-January time period, which means that the difference for this period with/without TROPOMI data is negligible. Therefore, I believe it is more meaningful to present &

report these biases for the 28 January – 3 May period, as done also in Figure 13.

We still want to show the biases for the complete period in Table 3 and 4 (were table 2 and3). However, as we now also have data after May we have split the statistics into the first part (26 Nov-3 May and the rest 11 June-30 Nov). For the average maps we now only show the periods Feb-April and Sep-Nov 2018.

15 P9,I.5: "26 November": Figure caption of Fig. 13 writes "28 January". Please check.

The Figure (now Fig10) now shows Feb-Apr and Sep-Nov and the caption has been adapted accordingly.

P9, I.10: "the impact would be larger": Yes, I agree, and this is worrying, particularly as the positive impact, seen so far, may no longer be the case. Could you please comment?

In our longer experiment the impact is still positive in the tropical troposphere, especially over West Africa, so we are not worried about this. Also, the new Figure 16 shows the small positive impact of TROPOMI (without OMI).

We have added in section 3.2.2:

"It seems the main advantage of assimilating TROPOMI into the CAMS system is the improvement of tropospheric ozone. This is due to the fact that MLS defines the stratosphere whereas OMI, GOME-2 and TROPOMI are also sensitive to the troposphere and add extra information here (see also Lefever et al., 2015).
Adding TROPOMI to the CAMS system fits the CAMS analysis better to independent tropospheric data."

P9, I.35 "observations"

Corrected

10

P9, I.39: "Loyola et al., 2019": the current data-stream appears preliminary, considering that the algorithm description has not been published yet. This is important to stress.

30 The TROPOMI data in this study are not preliminary. See our reply to the more general comment above.

P10, I.20 "solar elevation": this seems only partly an appropriate predictor. Would the CAMS system be flexible enough to add different predictors in the system for its bias correction scheme, such as albedo?

The CAMS bias correction scheme is flexible enough to work with other parameters (e.g. for IASI CO retrievals thermal contrast between surface and lowest model layer is one of the predictors), but experiments would be needed to test this.

P10, I.34 "to add redundancy", This is actually not tested, please see my comments above. Please

comment.

35

See our comment above regarding the new Figure 16.

Figure 10: Figure quality appears insufficient for publication, and legend seems incomplete. Please consider 40 improved legibility.

The quality of the Figure (now Figure 7) has been improved.

**Reviewer 2:**

The paper describes the outcomes of the assimilation tests in the CAMS system with Tropomi TCO3 v1.0.0 data. The paper properly explains the methods as well the input data used in this work. Presentation is clear and well structured. Proper acknowledgements are given to all authors as well as other data providers/sources.

The paper describes how CAMS assimilation system may be used to monitor bias as well as other measurement characteristics of a new instrument and thus, show how CAMS system may be used as an independent source of quality control. This may benefit both algorithm developers as well as data users.

The basic idea behind this work is not novel because the CAMS assimilation system has existed quite long time already. However, the Tropomi data itself is very interesting and can be seen as a step forward for the new instruments in future, like onboard EPS-SG platforms. The Tropomi instrument is in the afternoon orbit and thus, fulfils nicely data retrieved with morning instruments, like GOME-2 and other instruments in morning orbits. Thus, the topic of this paper is interesting indeed.

The paper show results over period 26th November to 3rd May 2018. Most of the figures show averaged values over the whole data period. However, the figures (like figure 7) indicate problems in L1 and L2 in December-January, which may be due to the early commissioning phase. Thus, the first data samples may not be at the same quality level than in Feb-April, which may have an effect to information content of the images showing averages of the whole period. Have you checked this? Thus, it would be interesting and useful to see a little bit more information about assimilation and control fields at certain fixed time steps.

20 We have increased the data period from 26 Nov 2017 to 30 Nov 2018 and now show timeseries for the whole period and averages for the periods February to April and September to October 2018 (July-September for comparison with IAGOS because of data availability). Thus the early problem periods in Dec 2017 and Jan 2018 do not affect the averages any more.

Furthermore, there are some other concerns at the same time. The first one is the length of the assimilation period from about December to April with several larger data caps. The data of about three months may not be well representative over the seasonal cycles, for example. Therefore, it is difficult to make conclusions beside February-April for the rest of the year.

**Period is now from 26 Nov 2017 to 30 Nov 2018.**

The second concern is about the TCO3 version v1.0.0. The current version is already 1.01.02 with improved OCRA and ROCINN as described by Loyola et al 2018. The retrieval algorithms for Tropomi are under fast development right now and thus, the results shown in this paper may be somewhat outdated already by now.

There are no big algorithm changes between the early data v1.0.0. and the current data v1.1.2, thus using the early data is meaningful. We do see some problems with the data during the commissioning phase but by extending the timeseries until the end of November 2019 we now use data from v1.0.0, v1.1.1, v1.1.2 giving more validity to the results.

The current version of the data-stream is not preliminary. The retrieval was well documented in the ATBD (http://www.tropomi.eu/documents/atbd , Jan 2019), which was last updated in October 2018. However, due to other priorities a respective manuscript has not been published yet.

S5P/TROPOMI Total Ozone ATBD Rob Spurr, Diego Loyola, Michel Van Roozendael, Christophe Lerot, Klaus-Peter Heue, Jian Xu: S5P-L2-DLR-ATBD-400A, issue 1.6, 2018-10-17

**We have added in section 2.2:**

35

"For the work in this paper we use NRT TROPOMI TCO3 data (Loyola et al.; 2019b). These include TROPOMI data (V1.0.0) for the period 26 November 2017 to 3 May 2018 V1.0.0 that were reprocessed with the NRT algorithm

and NRT TROPOMI data V1.0.0-V1.1.2 for the period 11 June to 30 November 2018 (see Table 2). No data were acquired at ECMWF from 4 May to 10 June 2018 for technical reasons. The TROPOMI TCO3 retrieval is based on the GDP 4.x algorithm original developed for GOME (van Roozendael et al., 2006), adapted to SCIAMACHY (Lerot et al., 2009) and further improved for GOME-2 (Loyola et al., 2011; Hao et al., 2014). The major TCO3 algorithm

- 5 updates for TROPOMI compared to the heritage algorithms used for GOME-2 are the more precise treatment of clouds as scattering layers (Loyola et al., 2018), an optimized wavelength for the calculation of air mass factors (328.2 nm instead of 325.5 nm), better a-priori ozone profile information (including the tropospheric climatology by Ziemke et al. (2011)) and a destriping correction. This destriping correction was introduced because total vertical ozone columns showed small striping structures. The correction factor is based on the ratio between the
- 10 mean for individual rows and the mean for all rows over a certain region and period. We averaged the total columns within the tropics for January to April 2018 for both all 450 rows individually and over all 450 rows, resulting in an array of 450 numbers, ranging between 0.99 and 1.015. Multiplying the VCD with the correction factor changes the result by about ±1%. The correction factor has been rechecked but no update seemed necessary up to now. The TROPOMI retrieval is described in the S5P/TROPOMI Total Ozone ATBD (Spurr et al., 2019)."
- 15 **2018)**."

Page 9, line 9-10 and page 10, lines 34-35: It's mentioned that if no other data is available, the effect of Tropomi data would be larger. This seems to be a justified conclusion indeed. Furthermore, according the paper, the main reason to assimilate Tropomi TOC3 v 1.0.0 and not to wait more mature data version is that the assimilation of Tropomi data as soon as possible would be beneficial in case of failure of older instruments.

20 However, to evaluate this properly, there should be assimilation tests where some other instruments are removed from the system. Otherwise, it may be difficult to support the argument because it's difficult to predict how CAMS model behaves when some other instruments are removed. I'm not expert in CAMS data assimilation and thus, some more evidence could be presented.

We have run 2 more experiments for some of the period (until the end of April 2018), one where TROPOMI was
assimilated and OMI was removed and one where both TROPOMI and OMI were removed. The results are shown in the new Fig 16. We have added:

"On the whole, the impact of the TROPOMI assimilation in the CAMS system is relatively small because the CAMS analysis is already well constrained by the other O3 data sets that are assimilated routinely which are a combination of TCO3 data (OMI, GOME-2AB), O3 layers (SBUV/2, OMPS) and O3 profiles (MLS) (see Table 1). If

- 30 no other O3 data were available and only TROPOMI TCO3 data were assimilated the impact on the CAMS O3 analysis would be larger. To confirm that TROPOMI could serve as a good replacement if one of the older TCO3 instruments (OMI, GOME-2AB) failed two further experiments were run for the period 26 November 2017 to 30 April 2018: one mimicking the configuration of CTRL, but without OMI (CTRL-OMI) and the other mimicking the configuration of ASSIM without OMI (ASSIM-OMI). Compared to ozonesondes and IAGOS data the differences
- 35 between these experiments and ASSIM and CTRL are very small indeed. The largest differences between the four experiments are found in the SH midlatitudes when compared with ozone sondes (Figure 16a) and over West African airports compared with IAGOS (Figure 16b). Even here, the differences between ASSIM and ASSIM-OMI are small and both fit the independent observations better than CTRL and CTRL-OMI. There is even a sign that removing OMI leads to a small improvement in the fit to IAGOS over West Africa. In all other areas the
- 40 differences between the experiments with and without OMI were negligible when compared to sondes or IAGOS. These findings agree with results from longer observation system experiments that were carried out with the CAMS system for the years 2013 and 2014 in a different context (not shown) which showed only small changes to the CAMS O3 analysis if one of the TCO3 instruments was removed confirming that the CAMS analysis is well constrained and that there is some redundancy in the system. We are therefore confident that TROPOMI will be
- 45 able to counterbalance the loss of one of the older TCO3 instruments. Removing MLS O3 profiles has a much larger (negative) impact on the CAMS O3 analysis (e.g. Flemming et al., 2011) and TROPOMI would not be able to replace the MLS profiles as it does not provided data with similar vertical resolution."

Several paragraphs/sentences associated to certain figures (like 4, 8, 11 and 13 etc) are very short. Thus, the analysis (text) and the figures don't seem to be in balance. Therefore, I would suggest that the authors should

reconsider the structure of the paper and put some of the paragraphs together, perhaps. Also, it could be useful to reconsider if all of those figures are necessary.

We have restructured the paper because of the longer time period and have also removed some of the Figures (i.e. previous Fig 2 and 3 are now combined in new Fig 3, Fig 6 is removed as the information is given in Table 3

6 (was table 2), Fig. 7 is removed and instead we show zonal mean hovmoeller plots of TROPOMI and differences with OMI and GOME-2Ab in the new Fig2, old Fig 8 is removed, Scatter plots against cloud cover and cloud top pressure have been removed. We also tried to put some paragraphs together.

Detailed comments:

Table 1: What is OMPS data version?

10 V1r0. This has been added to the table.

Page 3, line 20: It is difficult to see this as long term monitoring because the actual data period is about three months.

The period now covers a year from 26 Nov 2017 to 30 Nov 2018.

Page 4, line 8-10: A very short second paragraph in 2.1. Is it possible to merge this with the first paragraph in 15 2.1?

Done.

30

Page 6, line 18: "..is mainly lower..". This seems to be too general conclusion because the high max areas reach the latitude 65. There seems to be clear land-sea separation.

We have rephrased this: "The Figure shows that there are large differences between TROPOMI and GOME-2AB polewards of about 60° which seem to be mainly negative over ice and sea and positive over land."

Page 6, line 41: It would be better to use the version with the new climatology in this study.

This does not exist yet. It is planned to create a climatology from S5P data when a long enough record is available.

Page 7, line 1: ". . .small negative departures elsewhere. . .". There seems to be quite large negative departure over Antarctica but it's not mentioned here or before?

We are mentioning this now when looking at the new Figure 5: "During FMA (Figure 5b) TROPOMI is lower than the CAMS analysis south of  $60^{\circ}$ S and over land or snow/ice north of  $60^{\circ}$ N."

Figure 7, tables and texts: The latitudes are not consistent through the text. In fig 7 they are -70 – +70 whereas in tables 2 and 3 as -90 - +90. Furthermore, the active assimilation was done only within the latitude band of -60 - +60 and thus, the results in the Fig 7 could be different if the latitudes were restricted accordingly. Thus, the reason for different latitude bands should be explained (some are clear, some are not so).

Sorry for the confusion. We have removed Fig 7, so this is not an issue any more. We have also recalculated the statistics in Tables 2 and 3 (now Tables 3 and 4) for 5 latitude bands 90-60N, 60-30N, 30N-30S, 30-60S and 60-90S to bring them in line with the fact that we assimilate the data between 60N and 60S and to be able to
 calculate separate statistics for the polar regions where there are larger differences.

Page 7, line 30: "Small" is subjective term here. For example, 55N, the departure is about 10 DU. In general, to use 60N as a separation seems to be a little bit problematic because in several figures the actual separation could be 50N or 55N.

Figure 7 has been removed from the paper.

40 Figure 11 and Page 8, line 31-35: Is the swath angle dependency seen in tropics in the fig 11b?

No, this must have been an artefact of the colour scale and the averaging time period. We do not show the mean bias correction plots any more because they are not really meaningful as the bias correction changes with time. We still show the zonal mean timeseries of the bias correction.

Fig 12 b: Perhaps scale from -5 to 5 DU with 0.5 DU tics could work better.

5 We think the scale works well for the new averaging periods Feb-April and Sep-Oct 2018 and have kept it.

Page 9, line 33: Interesting to see this clear improvement

Yes, this is a nice result and is also seen in the new plots for July-Sept 2018. We stress this improvement more clearly in the text now.

**10**

**Differences between revised and original manuscript:**

**Monitoring and assimilation tests with TROPOMI data in the CAMS system: Near-real time total column ozone**

Antje Inness1, Johannes Flemming1, Klaus-Peter Heue2, Christophe Lerot3, Diego Loyola2, Roberto Ribas1, Pieter Valks2, Michel van Roozendael3, Jian Xu2 and Walter Zimmer2

1ECMWF, Shinfield Park, Reading, RG2 9AU, UK

2German Aerospace Centre (DLR), Remote Sensing Technology Institute, Oberpfaffenhofen, 82234 Wessling, Germany 3BIRA-IASB, Brussels, Belgium

Correspondence to: Antje Inness (a.inness@ecmwf.int)

**Abstract**

[revised manuscript text omitted]

Despite the small impact of TROPOMI TCO3 in the CAMS analysis it will be beneficial to include the TROPOMI TCO3 NRT data actively in the operational NRT CAMS analysis after more tests. This will add some redundancy and resilience in the system and will allow us to use a more robust observation system in case some of the other older instruments, whose retrievals are currently assimilated by CAMS, stop working.¶

[revised manuscript text omitted]

Figure 3 shows difference plots of TROPOMI and the other retrievals and illustrates that there are pronounced differences between the data sets that show up less clearly when looking at the absolute fields in Fig. 2. TROPOMI is higher than the other three retrievals in the NH south of 60°N, with positive differences of up to 60 DU in places between about 40-60°N. Poleward of that, TROPOMI is mainly lower than GOME-2AB while there are positive and negative differences compared to OMI. In the southern hemisphere (SH), the differences are smaller and mainly negative between 0-60°S, but larger negative differences are found over Antarctica. Because the differences show similar structures over Antarctica for OMI and GOME-2AB they are likely to point to problems with the TROPOMI retrievals rather than the other datasets. ¶

Figure 4 shows maps of the standard deviation of the four TCO3 retrievals over the period from 26 November 2017 to 3 May 2018. All retrievals show the same features with highest variability in the northern Extratropics and lowest variability in the Tropics.¶

Figure 5 shows mean analysis departures from the four TCO3 retrievals. Because the GOME-2AB and OMI retrievals are actively assimilated in the CAMS system and the analysis is drawing to the data their analysis departures are generally smaller than TROPOMI's. OMI has larger analysis departures than GOME-2AB with negative departures in the high Arctic and positive departures over the Tropics and over sea south of about 60°S. This might point to a problem with the OMI retrievals, which is not completely removed by the bias correction during the analysis. TROPOMI shows larger analysis departures than the other three retrievals, partly because the data are not being assimilated, but also because of issues with the TROPOMI NRT TCO3 retrievals. These problems show up better in the departure plots than in the plot of absolute values (Fig. 2). 
[revised manuscript text omitted]

Deleted: The positive departures north of 40°N can also be seen in histograms of departures (Fig. 6) which show a positive mean bias with respect to CAMS of 0.35±12.8 DU in the NH with a tail of large positive departures, a mean value of 0.69±3.54 DU in the Tropics and a negative mean value of -2.83±7.42 DU in the SH with a long tail of negative departures. The mean bias in the NH is small because the positive bias south of about 60°N and the negative one north of 60°N compensate. This is illustrated by the large standard deviation in the NH. Table 2 lists the mean biases and standard deviations from all four TCO3 retrievals and shows that the biases of TROPOMI are of similar magnitude to the biases of the other instruments. The standard deviation of the TROPOMI departures is larger than GOME-2AB's, but smaller than OMI's in the NH and similar to OMI's in the

Figure 7 shows timeseries of daily mean area averaged TROPOMI TCO3 departures, observation and analysis values as well as th number of observations from 26 November 2017 to 3 May 2018 for the NH, the Tropics and SH. The Figure shows that there are long periods without TROPOMI data in December 2017 and January 2018 when the instrument was undergoing calibration activities and a shorter period without data at the beginning of March 2018. The S5P satellite was still in its commissioning phase until 24 April 2018 which led to interruptions of the data availability. From mid-February onwards the number of data is more stable and the departures are smoother. The departures are positive in the NH and Tropics and negative in the SH. In the NH, the TROPOMI analysis departures are actually larger than the first-guess departures from mid-February onwards, showing that the assimilation of GOME-2AB and OMI adds information to the analysis that is contrary to the TROPOMI data. In the Tropics, the TROPOMI analysis departures are generally smaller than the first-guess departures illustrating that the data are more consistent with the other O3 data used in the analysis as the assimilation of the other data improves the fit of the analysis to the (not assimilated) TROPOMI data. In the SH, the TROPOMI firstguess and analysis departures are similar. Comparing the magnitude of the departures with the observation and analysis values shows that in relative terms the area averaged TROPOMI departures are small. They are less than 1% in the NH and SH and less than 0.5% in the Tropics.¶

Figure 8 shows a timeseries of zonal mean daily analysis departures from TROPOMI and illustrates that the evolution of the departures with time is quite stable between about 60%-60% where departures are small, but that there are larger variations (and larger departure)

**Deleted: 3 May**

| Deleted: , cloud cover, cloud top pressure                                                            |
|-------------------------------------------------------------------------------------------------------|
| Deleted: The                                                                                          |
| Deleted: 50                                                                                           |
| Deleted: 40                                                                                           |
| Deleted: poleward                                                                                     |
| Deleted: is in agreement                                                                              |
| Deleted: 5                                                                                            |
| Deleted: 8                                                                                            |
| Deleted: Furthermore                                                                                  |
| Deleted: small or slightly positive departures for solar elevations greater than about 25° and |
| Deleted: lower                                                                                        |
| Deleted: . There is no obvious                                                                        |
| Deleted: of the departures on cloud cover or cloud top pressure, but                           |
| Deleted: departures                                                                                   |
| Deleted: left                                                                                         |
| Deleted: This                                                                                         |
| Deleted: indicates possible limitations                                                               |
| Deleted: TROPOMI TCO3 NRT destriping correction.                                                      |

**3.2 Assimilation tests with TROPOMI TCO3 NRT data**

We showed in section 3.1 that the TROPOMI TCO3 data are of good quality over large parts of the globe, but that there are some issues at high latitudes and low solar elevations, especially in FMA. The biases we observe outside those regions are of similar magnitude to the biases of the other total column data sets assimilated in CAMS (see Table 3) and we therefore do not expect any problems with the assimilation of TROPOMI NRT TCO3 if we bias correct the data and blacklist them appropriately. Hence, assimilation tests are carried out with the TROPOMI NRT TCO3 data for the period 26 November 2017 to 30 Nov 2018, blacklisting them for solar elevations less than 10° and poleward of 60°. Restricting the assimilated data between 60°S and 60°N excludes the "ozone –hole" observation in these tests. Variational bias correction is applied to the data in the same way as it is used for the other TCO3 data, i.e. with solar elevation and a global constant as predictors. The choice of these bias correction parameters can be altered in the future if needed.

**3.2.1 Impact of the TROPOMI assimilation**

Figure 2 shows timeseries of global mean weekly averaged\_TROPOMI, OMI and GOME-2AB TCO3 departures, bias correction, standard deviation of departures and number of observations between 26 November 2017 and 30 November 2018 for 'used data', i.e. the data that fulfil the blacklist criteria and quality checks listed in Table 1 and pass the variational quality

15 control and first-guess checks applied by the IFS (see section 2.1). The figure shows that the TROPOMI bias correction successfully removes the biases between the data and the model, so that the bias corrected analysis departures are small. The bias correction calculates maximum values of about 1 DU in the global mean with the largest positive values between June and August and the largest negative values in November 2018. The magnitude of the global mean bias correction that is applied to TROPOMI is smaller than that of the other three TCO3 retrievals. Figure 7 shows that the analysis is drawing to the

20 TROPOMI data (and the other three datasets), i.e. analysis departures are smaller than the first-guess departures and the standard deviation of the departures is reduced. About 2.4 million TROPOMI observations are used every week which is 10x as many observations as from OMI, 5x as many as from GOME-2A and 3x as many as from GOME-2Br

30

35

5

10

Figure &a shows a timeseries of the zonal mean weekly averaged bias correction that is applied to TROPOMI data for the period 26 November 2017 to 30 November 2018. The Figure illustrates how the bias correction changes with time as it adapts to the data and that the mean bias-corrected TROPOMI analysis departures for FMA (Figure 8b) and SON (Figure 8c) are small compared to CTRL (Figure 5b and c) as the analysis is drawing to the TROPOMI data. Some larger positive departures for remain over land in the NH in FMA where observation outliers are given less weight by the analysis.

Figure 2 shows the mean TCO3 fields from ASSIM for FMA and SON 2018 as well as the absolute and relative differences between ASSIM and CTRL. It illustrates that the impact of the TROPOMI assimilation in relative terms is small with relative differences of less than 2% everywhere and less than 1% in most areas. The absolute differences are largest over land in the

40 NH in FMA with ASSIM up to 10 DU higher than CTRL. However, the absolute TCO3 values are also largest then. Positive differences are also found in an area stretching from South America over the Atlantic to Africa in FMA and SON and in small bands around 60°S. In most other areas, the differences are below -2 DU and negative.

| Deleted: 2                                                                                                                                                                                                    |    |
|---------------------------------------------------------------------------------------------------------------------------------------------------------------------------------------------------------------|----|
| Deleted: (V1.0.0)                                                                                                                                                                                             |    |
| Deleted: 3 May                                                                                                                                                                                                |    |
| Deleted: 10                                                                                                                                                                                                   |    |
| Deleted: daily                                                                                                                                                                                                |    |
| Deleted: 3 May                                                                                                                                                                                                |    |
| Deleted: check                                                                                                                                                                                                |    |
| Deleted: bias                                                                                                                                                                                                 |    |
| Deleted: and their standard deviations                                                                                                                                                                        |    |
| Deleted: has                                                                                                                                                                                                  |    |
| Deleted: 0.5                                                                                                                                                                                                  |    |
| Deleted: after March                                                                                                                                                                                          |    |
| Deleted: smaller or small                                                                                                                                                                                     |    |
| Deleted: before then.                                                                                                                                                                                         |    |
| Deleted: The figure                                                                                                                                                                                           |    |
| Deleted: We also see how many more data                                                                                                                                                                       |    |
| Deleted: from TROPOMI than                                                                                                                                                                                    |    |
| Deleted: : about 360,000 per day compared to 36,000, 75,000 and 110,0000 respectively                                                                                                                         | 1  |
| Deleted: 3                                                                                                                                                                                                    |    |
| Deleted: from ASSIM                                                                                                                                                                                           |    |
| Deleted: 2                                                                                                                                                                                                    |    |
| Deleted: In the Tropics, the bias and standard deviation of OMI
and GOME-2AB are also smaller in ASSIM than in CTRL because
of the additional constraint from the TROPOMI assimilation.          |    |
| Deleted: OMI and GOME-2A TCO3                                                                                                                                                                                 |    |
| Deleted: the NH and SH in ASSIM, but for GOME-2B it is                                                                                                                                                        |    |
| Deleted: all areas. The size of the TROPOMI                                                                                                                                                                   |    |
| Deleted: in Table 3 is similar to GOME-2AB's and smaller than OMI's                                                                                                                                    |    |
| Deleted: 11 shows                                                                                                                                                                                             |    |
| Deleted: the period 26 November to 3 May 2018                                                                                                                                                                 |    |
| Deleted: (                                                                                                                                                                                                    |    |
| Deleted: Fig.5) because                                                                                                                                                                                       |    |
| Deleted: The figure also shows that the mean bias correction for TROPOMI is positive at high latitudes and negative in the Tropics and that it changes with time to adjust to the changes in the data. |    |
| Deleted: 12                                                                                                                                                                                                   |    |
| Deleted: CTRL                                                                                                                                                                                                 |    |
| Deleted: their                                                                                                                                                                                                |    |
| Deleted: averaged over the period 26 November 2017 to 3 May 2018.                                                                                                                                      |    |
| Deleted: (both in absolute and                                                                                                                                                                                |    |
| Deleted: )                                                                                                                                                                                                    |    |
| Deleted: . Absolute differences are less than 1 DU over most of t Globe                                                                                                                                | he |
| Deleted: maximum                                                                                                                                                                                              |    |
| Deleted: DU around 50°N and over Africa, where ASSIM has larger TCO3 values than CTRL. In relative terms, the differences (                                                                            |    |
| Deleted: (                                                                                                                                                                                                    |    |
| Deleted: ) with negative relative                                                                                                                                                                             |    |
| Deleted: between 20-50°S, 20-40°N and                                                                                                                                                                         |    |
| Deleted: Pacific between 50°S and 40°N                                                                                                                                                                        | _  |
|                                                                                                                                                                                                               |    |

Figure 10 shows cross sections of zonal mean relative O3 mixing ratio differences from ASSIM minus CTRL averaged over FMA and SON. Again, the impact of TROPOMI assimilation is small with the largest relative differences found in the troposphere. Here the TROPOMI data act to lower the ozone values in ASSIM in the zonal mean. In FMA the impact is less

5 than 1% everywhere. In SON, the differences in the troposphere are slightly larger and reach values of up to -3% near the surface in NH midlatitudes and over the South Pole. Note that no TROPOMI data were assimilated south of 60°S so the changes seen here come from transport. Also, note that the absolute O3 values in the lower troposphere over the Antarctic are small.

**3.2.2 Validation with independent observations**

- 10 To assess if the assimilation of TROPOMI TCO3 retrievals improves or degrades the CAMS analysis, the O3 fields from ASSIM and CTRL are compared with independent observations. We use for comparison the following datasets. (1) Brewer spectrometer measurements obtained from the World Ozone and Ultraviolet Radiation Data Centre (WOUDC). The Brewer data are well calibrated with a precision of 1% (Basher, 1982). (2) Ozone sonde data from a variety of data centres: WOUDC, Southern Hemisphere ADditional OZonesondes (SHADOZ), Network for the Detection of Atmospheric Composition Change
- 15 (NDACC), and campaigns for the Determination of Stratospheric Polar Ozone Losses (MATCH). The precision of electrochemical concentration cell (ECC) ozone sondes is on the order of ±5% in the range between 200 and 10 hPa, between -14% and +6% above 10 hPa, and between -7% and +17% below 200 hPa (Komhyr et al., 1995). Larger errors are found in the presence of steep gradients and where the ozone amount is low. The same order of precision was found by Steinbrecht et al. (1998) for Brewer–Mast sondes. (3) Ozone profiles from instruments mounted on commercial aircraft from the In-service
- 20 Aircraft for a Global Observing System (IAGOS). The IAGOS ozone data have a detection limit of 2 ppbv and a precision of ± (2 ppbv + 2 %) (Marenco et al.,1998). (4) Ground-based data from the World Meteorological Organisation's Global Atmosphere Watch (GAW) surface observation network (e.g., Oltmans and Levy, 1994; Novelli and Masarie, 2014). The GAW observations represent the global background away from the main polluted areas. GAW O3 data have a precision of ±1 ppbv (Novelli and Masarie; 2014).
- 25

40

Figure 11 shows timeseries of the weekly averaged TCO3 biases from ASSIM and CTRL against Brewer measurements averaged over the Globe and NH midlatitudes for the period 26 November 2017 to 30 November 2018. The Figure shows a generally good agreement of both experiments with the Brewer data with maximum biases of less than 6 DU. It confirms that the impact of the TROPOMI assimilation in the CAMS system is small with differences between ASSIM and CTRL of less
than 1 DU in the total column. Despite being small, the impact usually leads to an improved fit to the WOUDC data in ASSIM.

Compared with ozone sondes averaged over FMA (Figure 12) and SON (Figure 13) the impact in relative terms is also small. However, an improved fit to the data is seen in ASSIM in the Tropics during SON when the positive bias seen in CTRL is reduced. Ozone profiles from ASSIM and CTRL are also compared with IAGOS aircraft data (Figure 14). Because not many
IAGOS profiles were available during October and November 2018 we show FMA and July-September (JAS) 2018. In both seasons, we see a positive impact from the assimilation of TROPOMI TCO3 data over West African airports where the negative bias seen in CTRL is reduced when assimilating TROPOMI TCO3 data. This increase in tropospheric O3 agrees with the increased TCO3 seen in ASSIM over Africa in FMA and SON (Figure 9), but does not show up in the zonal mean cross sections (Figure 10).

[revised manuscript text omitted]

Due to the limitations of the current TROPOMI TCO3 NRT product that uses a OMI climatology for the surface properties, ozone data had to be blacklisted at high latitudes in this study. Future algorithm updates dealing with a better treatment of the surface albedo (Loyola et al., 2019a) will improve the retrieval quality at high latitudes and should allow the data to be used up to the poles. The V1.1.2 data used after 8 August 2018 already show smaller departures south of 60°S. Note that the TROPOMI TCO3 offline algorithm does not have the limitation due to the surface albedo climatology seen in the NRT product

because the surface albedo is fitted as part of the retrieval.

TROPOMI TCO3 NRT data were included passively in the operational CAMS system on 13 July 2018, the day the data were officially released by ESA, and have been monitored routinely by CAMS ever since (see https://atmosphere.copernicus.eu/charts/cams\_monitoring). Because of the small, but positive impact of the TROPOMI TCO3 assimilation on the CAMS ozone analysis shown in this paper it was decided to actively include the TROPOMI TCO3 NRT data in the operational NRT CAMS analysis, and the routine assimilation of the data in the operational CAMS analysis began on 4 December 2018.

**Author Contributions**

15

A. Inness carried out the experiments described in the paper, the validation of the resulting analysis fields and wrote the manuscript, R. Ribas set up the S5P processing chain at ECMWF which included coding and testing the BUFR converter needed to ingest the TROPOMI TCO3 data in the ECMWF data system, J. Flemming helped with the development of the IFS
chemistry module, D. Loyola, W. Zimmer, K.-P. Heue, J. Xu, P. Valks, C. Lerot and M. van Roozendael developed the TROPOMI TCO3 retrieval algorithm and the operational processing chain at DLR. All co-authors gave useful comments during the writing of the paper.

**Acknowledgements**

Thanks to the DLR colleagues Fabian Romahn and Mattia Pedergnana working on the operational UPAS system for generating TROPOMI TCO3 products and thanks to Maximilian Schwinger and the PDGS team at DLR responsible for the Sentinel-5 Precursor payload data ground segment. Thanks to Luke Jones for help with the plotting of ozone sondes, GAW and IAGOS data. Thanks to the data providers of the data assimilated in the CAMS reanalysis and the data used for the validation studies in this paper. The TROPOMI total ozone data are generated at DLR on behalf of EU/ESA. The GOME-2 total ozone data assimilated in CAMS are provided by DLR in the framework of the EUMETSAT AC-SAF project. The Copernicus

30 Atmosphere Monitoring Service is operated by the European Centre for Medium-Range Weather Forecasts on behalf of the European Commission as part of the Copernicus programme (http://copernicus.eu).

**References**

Andersson, E. and Järvinen, H.: Variational quality control. Q.J. Roy. Meteor. Soc., 125,697-722, 1999.

35 Basher, R. E.: Review of the Dobson spectrophotometer and its accuracy, Global Ozone Res. Monit. Proj., Rep. 13, World Meteor. Organ., Geneva, Switzerland, December, available at: http://www.esrl.noaa.gov/gmd/ozwv/dobson/papers/report13/ report13.html (last access: 3 February 2017), 1982. Deleted: However, the tests also illustrate that the assimilation of TROPOMI data in the CAMS system does not degrade the ozone analysis, suggesting it should be straight forward to activate the TROPOMI assimilation in the operational NRT CAMS analysis. It would indeed be beneficial to include the TROPOMI TCO3 NRT data actively in the operational NRT CAMS analysis soon, despite the small impact, to add redundancy and resilience and to have a more robust observation system in place if some of the other older instruments, whose retrievals are currently assimilated by CAMS, stop working. ¶

**Deleted: V1.0.0**

**Deleted: this**

**Deleted:** The paper illustrates the power of using a global assimilation system to monitor new satellite products, as it provides continuous global coverage, allows us to build up global and regional statistics quickly and can help to identify problems with the retrievals (e.g. biases against solar elevation, latitude, scan position, surface albedo dependencies, etc.) that might be more difficult to discover when comparing TROPOMI retrievals against sparse in-situ observations. ¶

| Ì | Deleted: NRT                                   |
|---|------------------------------------------------|
| Ì | Deleted: when they                             |
| Ì | Deleted: Assimilation tests with               |
| Ì | Deleted: data continue and it is expected that |
| Ì | Deleted: of TROPOMI                            |
| Ì | Deleted: TCO3                                  |
| ١ | Deleted: will begin soon.                      |

[revised manuscript text omitted]
/   | Data product       | Data provider/version       | Blacklist criteria / | VarBC           | Reference                                     |
|---------------|--------------------|-----------------------------|----------------------|-----------------|-----------------------------------------------|
| Satellite     |                    |                             | thinning             | predictors      |                                               |
| GOME-2/       | TCO3               | AC-SAF/ DLR GDP4.8          | QF>0                 | Solar elevation | Hao et al. (2014), Valks et al. (2017)        |
| Metop-A       |                    |                             | SOE<6°               | Global constant |                                               |
|               |                    |                             | Thinned to 0.5°x0.5° |                 |                                               |
| GOME-2/       | TCO3               | AC-SAF/ DLR GDP4.8          | QF>0                 | Solar elevation | Hao et al. (2014), Valks et al. (2017)        |
| Metop-B       |                    |                             | SOE<6°               | Global constant |                                               |
|               |                    |                             | Thinned to 0.5°x0.5° |                 |                                               |
| MLS/          | O3 profiles        | NASA V3.4                   | QF>0                 | Not applied     | Schwartz et al. (2015)                        |
| Aura          |                    |                             | No thinning          |                 |                                               |
| OMI/          | TCO3               | NASA V883                   | QF>0                 | Solar elevation | Liu et al. (2010)                             |
| Aura          |                    |                             | SOE<10°              | Global constant |                                               |
|               |                    |                             | Thinned to 0.5°x0.5° |                 |                                               |
| OMPS (nadir)/ | O3 partial columns | NOAA/ Eumetsat V1r0         | QF>0                 | Solar elevation | Flynn et al. (2014)                           |
| Suomi NNP     |                    |                             | SOE<10°              | Global constant |                                               |
|               |                    |                             | No thinning          |                 |                                               |
| SBUV/2/       | O3 partial columns | NOAA V8                     | QF>0                 | Not applied     | Bhartia et al. (1996), McPeters et al. (2013) |
| NOAA-19       |                    |                             | SOE<6°               |                 |                                               |
|               |                    |                             | No thinning          |                 |                                               |
| TROPOMI/      | TCO3               | ESA/ DLR                    | QF>0                 | Solar elevation | Loyola et al. (2019 a )                |
| Sentinel-5P   |                    | V1.0.0-V1.1.2 (see Table 2) | SOE<10°              | Global constant |                                               |
|               |                    |                             | Abs(LAT)<60°         |                 |                                               |
|               |                    |                             | Super-obbed to T511  |                 |                                               |

Formatted Table

 Table 1: O3 satellite retrievals used in this paper. QF= quality flag given by data providers, SOE= Solar Elevation, LAT: Latitude, VarBC: Variational bias correction. The blacklist criteria describe when data were not used.

5

| Period                      | Version number         | Algorithm                      | Description of changes   |  |
|-----------------------------|------------------------|--------------------------------|--------------------------|--|
| 20171126-20180503           | V1. 0 0  | Reprocessed with NRT algorithm | N/A (original algorithm) |  |
| 20180611-20180718           | V1.0.0          | NRT                     | N/A (original algorithm) |  |
| 20180718-20180808           | V1.1.1          | NRT                            | Minor bugfixes, no       |  |
|                             |                        |                                | algorithm changes.       |  |
|                             |                        |                                | QA_values introduced     |  |
| 20180808-20181130           | V1.1.2          | NRT                     | Bug fix to time variable |  |
| Table 2: Version numbers of | TROPOMI data used in t | his study.                     |                          |  |

| Instrument | Period  | 90°-60°N    | 60°-30°N   | 30°N-30°S | 30°-60°S   | 60°-90°S   |
|------------|---------|--------------------|-------------------|------------------|-------------------|-------------------|
| TROPOMI    | Nov-May | $-1.07 \pm 17.30$  | 2.10±9.47         | 0.06±3.83        | -0.05±4.95        | -6.81±7.32 |
| OMI | Nov-May | -2.85±8.11  | -2.70±7.99        | 0.18±7.45        | 0.83±7.35         | 2.17±6.42  |
| GOME-2A    | Nov-May | 0.81±6.35          | -0.60±5.88        | 0.06±3.14        | -0.10±3.40 | 0.86±3.38         |
| GOME-2B    | Nov-May | 0.29±6.25          | 0.42±6.14         | 0.16±2.97        | -0.19±3.33        | -0.46±3.31        |
| TROPOMI    | Jun-Nov | -1.46±10.40 | 0.31±6.00         | -0.47±3.88       | 0.82±10.10        | -2.39±6.99        |
| OMI | Jun-Nov | -1.57±7.22  | -1.63±7.77 | 0.48±7.15        | 0.73±7.95         | 2.07±6.65         |
| GOME-2A    | Jun-Nov | 0.19±5.11          | -0.25±4.51        | 0.02±3.55        | 0.24±5.08         | 0.67±3.56         |
| GOME-2B    | Jun-Nov | 0.68±4.80          | 0.53±4.45         | -0.18±3.27       | 0.04±5.08         | -0.29±3.65        |
| T 11 2 16  |         | 1 1 1 1 1 0 1      | TCO2 ( 1 1        |                  |                   | DILE (1 ( )       |

| Deleted: | Instrument          |
|----------|---------------------|
| Formatte | ed Table            |
| Deleted: | NH (20-90ºN)        |
| Deleted: | Tropics (20°S-20°N) |
| Deleted: | SH (20-90°S)        |
| Deleted: | TROPOMI (good data) |
| Deleted: | 35±12.8             |
| Deleted: | 0.69±3.54           |
| Deleted: | -2.83±7.42          |
| Deleted: | OMI (used data)     |
| Deleted: | -2.48±7.88          |
| Deleted: | 0.73±7.38           |
| Deleted: | 1.01±7.12           |
| Deleted: | GOME-2A (used data) |
| Deleted: | -0.15±5.64          |
| Deleted: | 0.16±2.97           |
| Deleted: | 0.24±3.45           |
| Deleted: | GOME-2B (used data) |
| Deleted: | 0.35±5.8            |
| Deleted: | 0.24±2.82           |
| Deleted: | -0.29±3.36          |
| Deleted: | Table 2             |
| Deleted: | ).                  |

Table 3: Mean bias and standard deviations of the TCO3 retrievals against the CAMS ozone analysis in DU from the control experiment (CTRL) for the periods 26 November 2017 to 3 May 2018 and 11 June to 30 November 2018. Green numbers mark 10

where the biases and standard deviations of the other TCO3 datasets are smaller than TROPOMI's, red marks where they are larger. Shown are 'good' data for TROPOMI and 'used' data for the other instruments.

| Instrument  | Period      | 90 °-60 °N  | 60°-30°N            | 30°N-30 °S        | 30°-60°S          | 60°- 90°S |
|-------------|-------------|--------------------|----------------------------|--------------------------|--------------------------|------------------|
| (used data) |             |                    |                            |                          |                          |                  |
| TROPOMI     | Nov-        | Not used           | 0 51 ±6 64   | 0 07 ±2 44 | 0.003±3.24               | Not used         |
|             | May  |                    |                            |                          |                          |                  |
| OMI         | Nov-        | - 3.19+8.09 | -3.34±8.25                 | 0 22 ±7 42 | 1 08+ 7 21 | 2.20±6.41        |
|             | May  |                    |                            |                          |                          |                  |
| GOME-2A     | Nov-        | 0.71±6.38          | -0 -90 ±5 88 | 0. 06±3.13        | 0. 02+3.56        | 0.86±3.38        |
|             | May  |                    |                            |                          |                          |                  |
| GOME-2B     | Nov-        | 0 20±6.28   | 0 14±6.27           | 0.20±3.06                | -0 13±343         |                  |
|             | May         |                    |                            |                          |                          | 0.47±3.31        |
| TROPOMI     | Jun- | Not used           | 0.07±4.09                  | 0.03±2.64                | 0.29±4.63                | Not used         |
|             | Nov         |                    |                            |                          |                          |                  |
| OMI  | Jun- | -1.76±7.22         | -1.82±7.87                 | 0.67±7.10                | 0.25±7.75                | 2.03±6.63        |
|             | Nov         |                    |                            |                          |                          |                  |
| GOME-2A     | Jun-        | 0.17±5.12          | -0.15±4.60          | 0.05±3.77                | 0.03±5.24                | 0.59±3.61        |
|             | Nov         |                    |                            |                          |                          |                  |
| GOME-2B     | Jun- | 0.71±4.80          | 0.68±4.67                  | -0.16±3.45               | -0.29±5.16               | =                |
|             | Nov         |                    |                            |                          |                          | 0.41±3.69        |

5 Table 4: Mean bias and standard deviations of the TCO3 retrievals against the CAMS ozone analysis in DU from the assimilation experiment (ASSIM for the periods 26 November 2017 to 3 May 2018 and 11 June to 30 November 2018 for 'used' data. Green numbers mark where the biases or standard deviations are smaller in ASSIM than in CTRL (Table 3), red numbers mark where they are larger.

| Deleted: NH (200º-60ºN)                                                                                                                      | (                       |
|----------------------------------------------------------------------------------------------------------------------------------------------|-------------------------|
| Inserted Cells                                                                                                                               |                         |
| Inserted Cells                                                                                                                               |                         |
| Deleted: Tropics (200°N-30°S-20°N)                                                                                                           |                         |
| Formatted Table                                                                                                                              |                         |
| Inserted Cells                                                                                                                               |                         |
| Deleted: SH (200º-90ºS)                                                                                                                      |
                    |
| Inserted Cells                                                                                                                               |                         |
| Deleted: 2911±6.01                                                                                                                           |                         |
| Deleted: 1747±2,20                                                                                                                           |                         |
| Deleted:05803±3.04                                                                                                                           |                         |
| Inserted Cells                                                                                                                               |                         |
| Deleted: 04                                                                                                                                  |                         |
| Formatted: Font color: Red                                                                                                                   |                         |
| Deleted: 2.74                                                                                                                                |                         |
| Deleted: 06                                                                                                                                  |                         |
| Inserted Cells                                                                                                                               |                         |
| Formatted: Font color: Red                                                                                                                   |                         |
| Deleted: 51                                                                                                                                  |                         |
| Formatted: Font color: Green                                                                                                                 |                         |
| Deleted: 36                                                                                                                                  |                         |
| Formatted: Font color: Red                                                                                                                   |                         |
| Formatted: Font color: Green                                                                                                                 |                         |
| Deleted: 21                                                                                                                                  |                         |
| Formatted: Font color: Green                                                                                                                 |                         |
| Inserted Cells                                                                                                                               |                         |
| Formatted: Font color: Red                                                                                                                   |                         |
| Deleted: 2670±5.73                                                                                                                           |                         |
| Deleted:006±2.97                                                                                                                             |                         |
| Formatted: Font color: Green                                                                                                                 |                         |
| Deleted: 346                                                                                                                                 |                         |
| Formatted: Font color: Red                                                                                                                   |                         |
| Deleted: 45                                                                                                                                  |                         |
| Formatted: Font color: Green                                                                                                                 |                         |
| Deleted: 287±5.86                                                                                                                            |                         |
| Formatted: Font color: Green                                                                                                                 |                         |
| Deleted: 083±2.88                                                                                                                            |                         |
| Inserted Cells                                                                                                                               |                         |
| Formatted: Font color: Green                                                                                                                 |                         |
| Deleted: 21                                                                                                                                  |                         |
| Formatted: Font color: Red                                                                                                                   |                         |
| Deleted: 38                                                                                                                                  |                         |
| Deleted: 3: Mean bias and standard deviations
retrievals against the CAMS ozone analysis in DU
assimilation experiment (ASSIM). | of the TCO3
from the |